# Alternative splicing in lung influences COVID-19 severity and respiratory diseases

Tomoko Nakanishi [1,2,3,4,5,13,14] ✉, Julian Willett [2,6,7,13], Yossi Farjoun[2,8], Richard J. Allen[9,10], Beatriz Guillen-Guio[9,10], Darin Adra[2], Sirui Zhou[1,6,7] & J. Brent Richards [2,8,11,12,14] ✉

Alternative splicing generates functional diversity in isoforms, impacting immune response to infection. Here, we evaluate the causal role of alternative splicing in COVID-19 severity and susceptibility by applying two-sample Mendelian randomization to *cis*-splicing quantitative trait loci and the results from COVID-19 Host Genetics Initiative. We identify that alternative splicing in lung, rather than total expression of *OAS1*, *ATP11A*, *DPP9* and *NPNT*, is associated with COVID-19 severity. *MUC1* and *PMF1* splicing is associated with COVID-19 susceptibility. Colocalization analyses support a shared genetic mechanism between COVID-19 severity with idiopathic pulmonary fibrosis at the *ATP11A* and *DPP9* loci, and with chronic obstructive lung diseases at the *NPNT* locus. Last, we show that *ATP11A, DPP9, NPNT*, and *MUC1* are highly expressed in lung alveolar epithelial cells, both in COVID-19 uninfected and infected samples. These findings clarify the importance of alternative splicing in lung for COVID-19 and respiratory diseases, providing isoform-based targets for drug discovery.

Despite the successful development of vaccines[1,2] and treatments[3,4], hospital admission for severe COVID-19 and long-term sequela of COVID-19 remain common[5,6]. COVID-19 is now a leading cause of death, accounting for more than 6 million deaths worldwide[7]. Thus, there is an ongoing need to identify mechanistic targets for therapeutic development to reduce the risk of severe COVID-19.

Using human genetics methods, several host factors have been identified to influence COVID-19 severity, including *OAS1*, type I interferon and chemokine genes[8–13]. Human genetics can provide new biological insights into disease pathogenesis, and therapeutic targets with evidence from human genetics enjoy an increased probability of

drug development success[14,15]. While prior studies have evaluated causal roles of both RNA expression and circulating proteins in COVID-19 outcomes[8,11,12], the causal role of alternative splicing has not been fully investigated.

Alternative splicing is an essential mechanism for generating functional diversity in the isoforms, through which multiple mRNA isoforms are produced from a single gene, often in tissue-specific patterns[16]. Alternative splicing has been implicated in immune response to infections in humans[16], and this might be the case for SARS-CoV-2 infection. The genetic determinants of alternative splicing may be identified using splicing quantitative trait loci (sQTL) studies[17],

[1]Department of Human Genetics, McGill University, Montréal, QC, Canada. [2]Lady Davis Institute, Jewish General Hospital, McGill University, Montréal, QC, Canada. [3]Kyoto-McGill International Collaborative Program in Genomic Medicine, Graduate School of Medicine, Kyoto University, Kyoto, Japan. [4]Department of Genome Informatics, Graduate School of Medicine, the University of Tokyo, Tokyo, Japan. [5]Research Fellow, Japan Society for the Promotion of Science, Tokyo, Japan. [6]Quantitative Life Sciences Program, McGill University, Montréal, Canada. [7]McGill Genome Centre, McGill University, Montréal, QC, Canada. [8]Five Prime Sciences Inc, Montréal, QC, Canada. [9]Department of Population Health Sciences, University of Leicester, Leicester, United Kingdom. [10]National Institute for Health Research, Leicester Respiratory Biomedical Research Centre, Glenfield Hospital, Leicester, UK. [11]Departments of Medicine, Human Genetics, Epidemiology and Biostatistics, McGill University, Montréal, QC, Canada. [12]Department of Twin Research, King's College London, London, UK. [13]These authors contributed equally: Tomoko Nakanishi, Julian Willett. [14]These authors jointly supervised this work: Tomoko Nakanishi, J. Brent Richards. ✉e-mail: tomoko.nakanishi@mail.mcgill.ca; brent.richards@mcgill.ca

where such studies indicate that splicing is under strong genetic control in humans and often has direct effects on protein isoforms. We have recently identified that a Neanderthal-introgressed isoform of OAS1, which is strongly regulated by an sQTL, protects against COVID-19 severity[8,18]. Given this evidence, we aimed to determine if alternative splicing could partially explain the variability in COVID-19 outcomes in humans at other genes.

In this study, we undertook two-sample Mendelian randomization (MR) and colocalization analyses to determine whether RNA splicing influences COVID-19 outcomes[19]. We first identified cis-sQTLs in lungs and whole blood, two relevant tissues that influence acute SARS-CoV-2 infection[20], from the GTEx Consortium v.8[17]. We then used MR to assess whether these RNA splicing events influence on COVID-19 outcomes using the GWASs from the COVID-19 Host Genetics Initiative release 7[10]. Next, all findings were assessed for colocalization to determine whether RNA splicing events and COVID-19 outcomes shared a common etiological genetic signal and that the MR results were not biased by linkage disequilibrium (LD). We also compared the effects of alternative RNA splicing to total RNA expression, and evaluated the expression of identified genes in the lung transcriptome. Finally, we evaluated whether such alternative splicing is a shared pathophysiological mechanism with other respiratory diseases.

## Results
### MR using cis-sQTLs, and colocalization analyses
The study design is illustrated in Fig. 1. We obtained sQTLs, which were quantified as normalized intron excision ratios, i.e., the proportion of reads supporting each alternatively excised intron, using LeafCutter[21] (with a multiple-testing correction threshold of a false discovery rate of 5% per tissue) in GTEx v.8[17]. We chose to examine only cis-sQTLs, which are more likely to act on local coding genes instead of through horizontally pleiotropic mechanisms, where such genes would have an effect on disease (here, COVID-19 outcomes) independently of the exposure (here, alternative splicing). We focused on two COVID-19-relevant tissues; lungs ($N = 452$ of European American ancestry) and whole blood ($N = 570$ of European American ancestry)[17]. We chose these tissues since pulmonary symptoms are the major determinant of hospital admission and blood immune cells in blood play a major role in host response to SARS-CoV-2[20].

A total of 5724 transcriptional splicing events for 4329 genes in lung and 3568 transcriptional splicing events for 2671 genes in whole blood contained conditionally independent cis-sQTLs that were also present the GWAS meta-analyses of European ancestry in COVID-19 Host Genetics Initiative[10] release 7 (https://www.covid19hg.org/results/r7/), which included results from the GenOMICC[11] study, but not the 23andMe study. We then undertook two-sample MR analyses using 5807 cis-sQTLs in lungs and 3658 cis-sQTLs in whole blood as genetic instruments for transcriptional splicing against three COVID-19 outcomes: (1) critical illness (defined as individuals experiencing death, mechanical ventilation, non-invasive ventilation, high-flow oxygen, or use of extracorporeal membrane oxygenation) owing to symptoms associated with laboratory-confirmed SARS-CoV-2 infection (13,769 cases and 1,072,442 controls); (2) hospitalization owing to symptoms associated with laboratory-confirmed SARS-CoV-2 infection (32,519 cases and 2,062,805 controls); and (3) reported SARS-CoV-2 infection defined as laboratory-confirmed SARS-CoV-2 infection, electronic health record, clinically confirmed COVID-19, or self-reported COVID-19, with or without symptoms of severity (122,616 cases and 2,475,240 controls). Our MR analyses used data from individuals of European ancestry to reduce the risk of bias from population stratification. We could not perform sex-stratified analyses due to the lack of the availability of the sex-stratified GWAS summary statistics.

MR analyses identified 43 transcriptional splicing events in lung and 26 transcriptional splicing events in whole blood which influenced

COVID-19 outcomes ($p < 1.8 \times 10^{-6}$, a Bonferroni-corrected p-value threshold which accounted for the number of tests performed [$N = 27,230$] with Type I error rate of 0.05, Methods, Supplementary Data 1, 2). We first replicated the association of COVID-19 outcomes with OAS1 splicing, using an updated version of COVID-19 HGI release 7, which provides a 10-fold increase in case sample size. The OAS1 sQTL, rs10774671:A > G, which increases the excision of the intron junction at chr12:112,917,700-112,919,389 [GRCh38] by 1.7 SD per one copy in lung (Fig. 2a, Supplementary Fig. 1A) and by 1.8 SD per one copy in whole blood, respectively, was associated with protection against all three adverse COVID-19 outcomes. The higher excision of the intron junction at chr12:112,917,700-112,919,389 corresponds to an increased level of the p46 isoform[22], a prenylated form of OAS1 with higher anti-viral activity than the p42 isoform[8,18,23] (Fig. 3, Supplementary Data 1,2). While rs10774671 is in LD with two coding variants, rs2660 ($r^2 = 0.97$) and rs1859330 ($r^2 = 0.87$), it is more likely that rs10774671 is the causal variant for the splicing effect, given that rs10774671 is a well-known splice-acceptor variant which has been functionally validated[22].

We additionally identified that ATP11A sQTL, rs12585036:T > C, which increases the excision of the intron junction at chr13:112,875,941-112,880,546 by 0.56 SD in lung (Fig. 2b, Supplementary Fig. 1B), was associated with protection against all three adverse COVID-19 outcomes with odds ratio [OR] per normalized intron excision ratio (the proportion of reads supporting each alternatively excised intron identified by LeafCutter[21]) of 0.78 (95%CI: 0.73–0.83, $p = 2.3 \times 10^{-16}$) for critical illness, OR of 0.84 (95%CI: 0.81–0.88, $p = 9.4 \times 10^{-17}$) for hospitalization, and OR of 0.96 (0.94–0.98, $p = 9.3 \times 10^{-6}$) for reported infection (Fig. 3, Supplementary Data 1). ATP11A has a single protein-coding transcript (ENST00000415301) that excises the targeted intron (chr13:112,875,941-112,880,546, Supplementary Fig. 1B).

We also found novel associations of DPP9 sQTL, rs12610495:G > A, which increases the excision of the intron junction at chr19:4,714,337-4,717,615 by 0.26 SD in lung (Fig. 2c, Supplementary Fig. 1C), with protection against all three adverse COVID-19 outcomes (OR: 0.39, 0.35–0.44, $p = 3.1 \times 10^{-51}$ for critical illness, OR: 0.56, 0.52–0.61, $p = 4.2 \times 10^{-41}$ for hospitalization, and OR: 0.82, 0.79–0.86, p = $1.8 \times 10^{-22}$ for reported infection, Fig. 3). DPP9 has a single protein-coding transcript (ENST00000599248) that excises the targeted intron (chr19:4,714,337-4,717,615, Supplementary Fig. 1C).

NPNT sQTL, rs34712979:A > G, which increases the excision of the intron junction at chr4:105,898,001-105,927,336 by 0.64 SD in lung (Fig. 2d, Supplementary Fig. 1D), was associated with increased risk of severe COVID-19 outcomes (critical illness and hospitalization) but with weak evidence of increased risk of SARS-CoV-2 infection (OR: 1.19, 1.13–1.25, $p = 1.2 \times 10^{-10}$ for critical illness, OR: 1.11, 1.07–1.15, $p = 2.2 \times 10^{-8}$ for hospitalization, and OR: 1.02, 1.00–1.03, $p = 4.9 \times 10^{-2}$ for reported infection, Fig. 3, Supplementary Data 1). The NPNT sQTL, rs34712979-A allele, creates a NAGNAG splice acceptor site, which results in additional in-frame AGT codon, coding for serine, at the 5' splice site of exon 2[24]. The rs34712979-A allele also serves as a protein-QTL that associates with decreased circulating levels of plasma NPNT[25-27], which could reflect the aptamer binding effect impacted by the alternative splicing.

Alternative splicing in the two genes in the gene cluster located in chr1 q21.3-q22, namely MUC1, and THBS3 in lung were also associated with reported SARS-CoV-2 infection, but not with COVID-19 severity phenotypes (Fig. 3, Supplementary Data 1). Given that the MUC1 and THBS3 sQTL variants are in high LD with each other (rs4072037:C > T and rs2066981:A > G; $r^2 = 0.98$ in European population of 1000 G), it was challenging to distinguish which of the two genes were more likely to be causal. Nevertheless, the MUC1 sQTL variant, rs4072037:C > T, is a recognized splice variant which influences the 3' splice site selection of exon 2, which leads to transcripts that have an alternative 27 bp

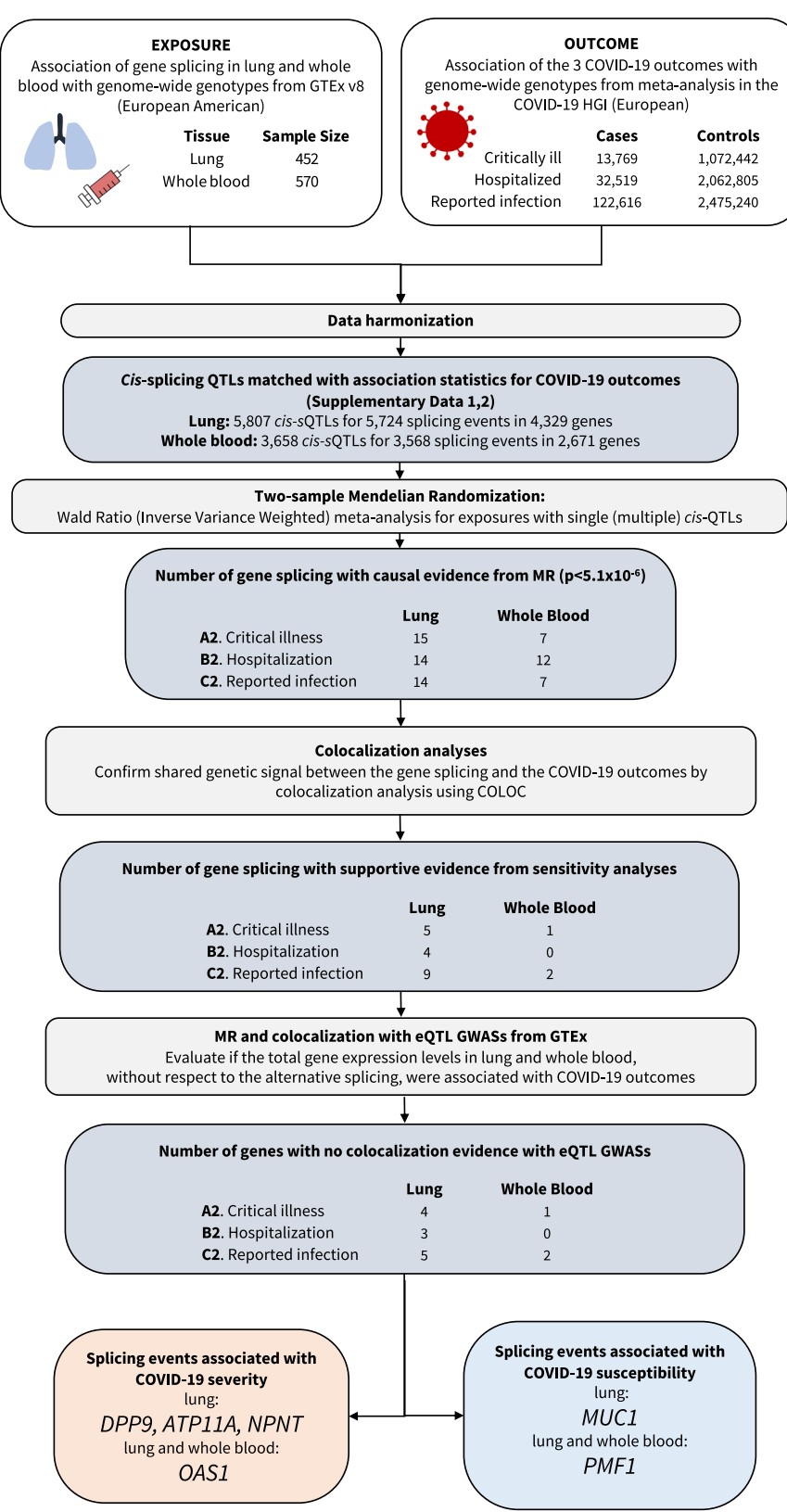

**Fig. 1 | Flow Diagram of Study Design.** MR Mendelian randomization.

intron retention event at the start of exon 2[28]. We demonstrated the rs4072037:T > C, which increases the excision of the intron junction at chr1:155,192,310–155,192,786 by 1.53 SD in lung (Fig. 2e, Supplementary Fig. 1E), was associated with increased risk of reported SARS-CoV-2 infection. Given the evidence of the splicing event of *MUC1*[28], it is

perhaps more likely that *MUC1* splicing influences SARS-CoV-2 infection.

*PMF1* sQTL, rs1052067:G > A, which increases the excision of the intron junction at chr1:156,233,728–156,236,349 by 1.4 SD in lung (Fig. 2f, Supplementary Fig. 1F) and by 1.5 SD in whole blood, was

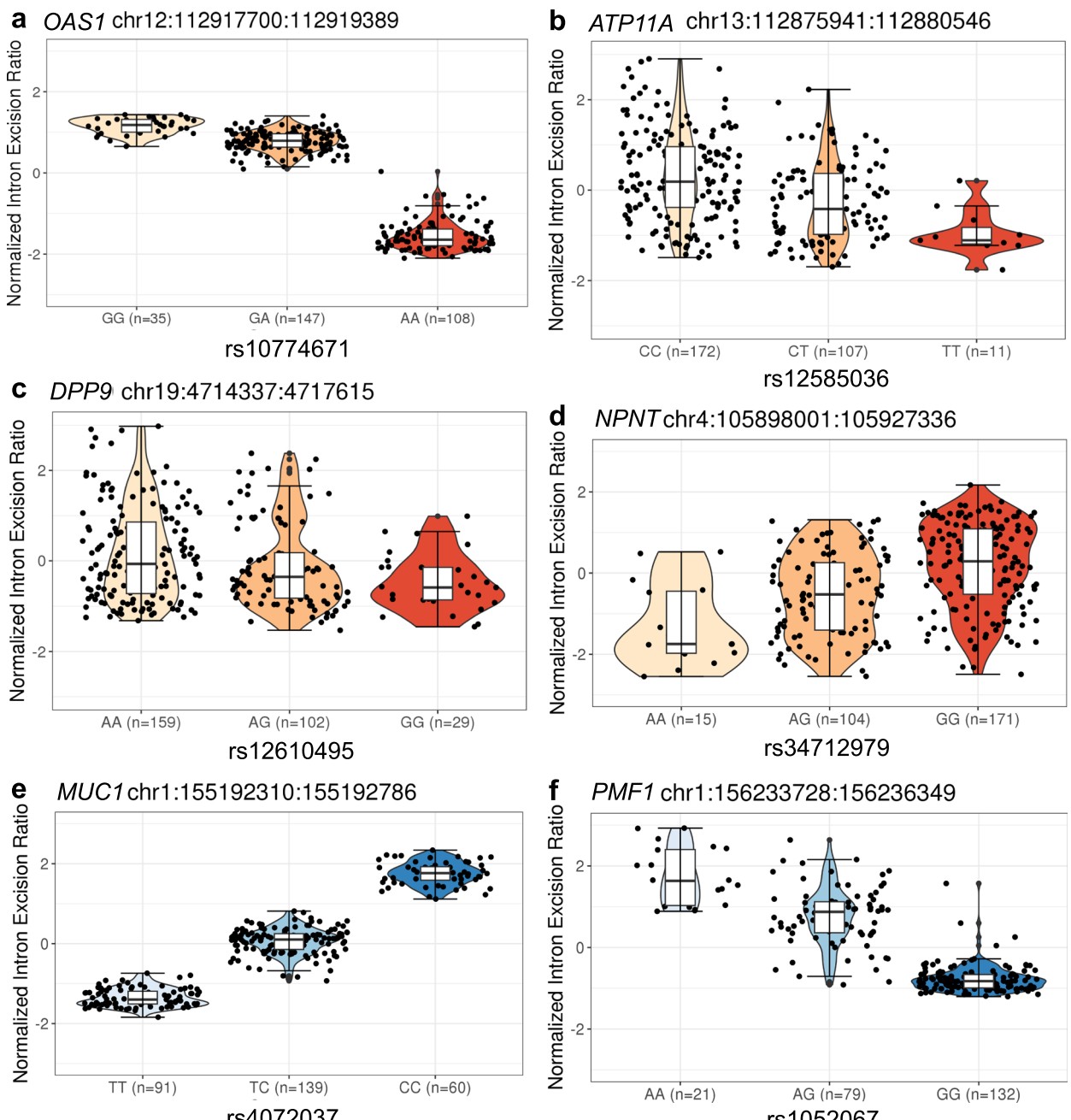

**Fig. 2 | The violin plots of normalized intron excision ratio stratified by sQTL genotypes. a** *OAS1*, **b** *ATP11A*, **c** *DPP9* and **d** *NPNT* show the sQTL genotypes in dark orange that are associated with COVID-19 severity. **e** *MUC1* and **f** *PMF1* show the sQTL genotypes in dark blue that are associated with SARS-CoV-2 reported infection. Normalized intron excision ratios were obtained from GTEx sQTL phenotype matrices (https://www.gtexportal.org/home/datasets). The genotypes were obtained from whole exome sequence data. Lower edge of the whisker: the lowest value within 1.5 * IQR of the hinge, lower hinge: 25% quantile, horizontal line contained within the box: median value, upper hinge: 75% quantile, the upper edge of the whisker: the highest value that is within 1.5 * IQR of the hinge.

associated with reduced risk of SARS-CoV-2 infection (OR: 0.98, 0.98–0.99, $p = 8.1 \times 10^{-7}$, both in lung, Fig. 3, Supplementary Data 1, and whole blood, Supplementary Data 2). rs1052067:G > A is a missense variant which replaces methionine with isoleucine at amino acid position 137 of the MANE transcript, ENST00000368277. This variant creates another transcript, ENST00000368279, with an alternative 61 bp intron retention event at the start of exon 4 (Supplementary Fig. 1F).

All above associations of alternative splicing, except for *MUC1* and *PMF1*, were more pronounced as the severity of the COVID-19 outcome increased (Fig. 3). We also applied MR for another hospitalization phenotype compared within COVID-19 cases ("hospitalization vs non-

hospitalization amongst individuals with laboratory-confirmed SARS-CoV-2 infection", which corresponds to B1 phenotype in COVID-19 HGI[10]). Here we used the data from all ancestries, as we did not have access to ancestry specific GWAS summaries for this phenotype. We confirmed the alternative splicing associations with *ATP11A* and *DPP9* (*ATP11A*: OR 0.88, 0.81–0.94, $p = 4.6 \times 10^{-4}$, and *DPP9*: OR 0.78, 0.66–0.93, $p = 4.9 \times 10^{-3}$) but not with *NPNT*, and *OAS1* (Supplementary Data 3).

**Colocalization analyses**

To test whether confounding due to linkage disequilibrium may have influenced the MR estimates, we tested the probability that RNA

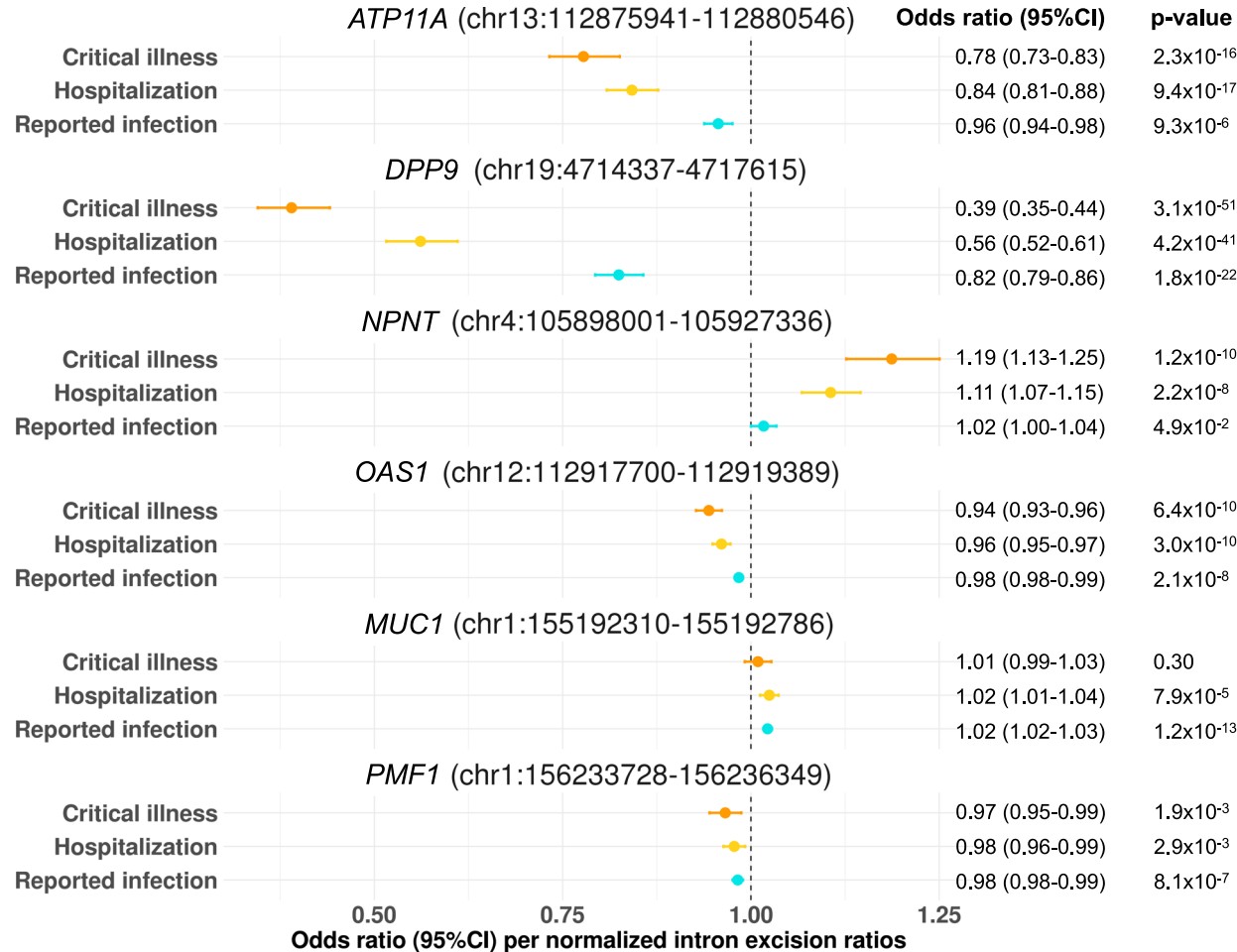

**Fig. 3 | MR estimates of the effect of RNA splicing at *ATP11A*, *DPP9*, *NPNT*, *OAS1*, *MUC1*, and *PMF1* in lung with COVID-19 outcomes.** Forest plot showing odds ratio and 95% confidence interval from two sample Mendelian Randomization analyses (two-sided). All significant results listed in Fig. 3. were estimated by Wald ratio. *P* values are unadjusted. Unit of exposure: standard deviation of intron excision ratios as quantified by LeafCutter. Centre: Odds ratio, error bar: 95% confidence interval (CI).

splicing and the COVID-19 outcomes shared a single causal signal using colocalization analyses, as implemented in coloc[29].

Amongst the MR-prioritized alternative splicing (Supplementary Data 1, 2), we selected the transcriptional splicing with high (>0.8) posterior probability[29] that the COVID-19 outcome shares a single causal signal with the alternative splicing event (referred to as hypothesis 4 in coloc[29], Supplementary Fig. 2). These colocalization analyses supported transcriptional splicing's role in nine genes (*ABO*, *ATP11A*, *DPP9*, *GBAP1*, *MUC1*, *NPNT*, *OAS1*, *PMF1*, and *THBS3*) in lung and two genes (*OAS1* and *PMF1*) in whole blood that influenced at least one COVID-19 outcome (Supplementary Data 4, 5).

The posterior probability that alternative splicing of *ATP11A* in lung and COVID-19 outcomes shared a single causal signal in the 1 Mb locus around the *cis*-sQTL was 1.00 for critical illness, 1.00 for hospitalization due to COVID-19, and 0.98 for reported infection (Supplementary Fig. 2A, Supplementary Data 4). Alternative splicing of *DPP9* also had high posterior probabilities of a shared single causal signal for critical illness (0.96), hospitalization (0.96), and reported infection (0.95) (Supplementary Fig. 2B, Supplementary Data 4). Alternative splicing of *NPNT* had a high posterior probability for critical illness (1.00), and hospitalization (1.00), but had a low posterior probability for reported infection (0.02) (Supplementary Fig. 2C, Supplementary Data 4). Lastly, alternative splicing of *MUC1 and PMF1* had high posterior probabilities for reported infection (1.00 and 0.95, respectively, Supplementary Fig. 2D, E, Supplementary Data 4).

### MR and colocalization with eQTL GWASs from GTEx

We next sought to determine if the genes identified through *cis*-sQTL studies also influenced COVID-19 outcomes through total gene expression, irrespective of splicing. To do so we used eQTL studies identified through Open Targets Genetics[30,31] (https://genetics.opentargets.org/, Supplementary Data 6). We performed MR and colocalization analyses using *cis*-expression quantitative trait loci (*cis*-eQTLs) in lung and whole blood from GTEx. In lung, the total expression of *ABO* and *GBAP1* had MR evidence with high colocalization (a posterior probability >0.8) for COVID-19 outcomes (all three COVID-19 outcomes with *ABO* and reported infection with *GBAP1*, Supplementary Data 7). In whole blood, the eQTLs of *ABO* had high colocalization with critical illness and hospitalization (Supplementary Data 8). For those genes, we could not clarify whether the associations with COVID-19 outcomes were driven by either total gene expression or the spliced isoform expression, or both.

### Influence of prioritized sQTLs on other diseases

We next searched for the effects of the sQTLs, which influence COVID-19 outcomes, on other diseases. To do so, we used the Open Targets Genetics[30,31] to identify diseases and disease-associated traits with its GWAS lead variants in LD ($r^2 > 0.80$) with COVID-19 associated sQTLs (Table 1). We found that the *DPP9* sQTL (rs12610495:G > A) was associated with decreased risk of idiopathic pulmonary fibrosis (IPF)[32] where the same allele confers the protection against COVID-19

**Table 1 | Colocalization analyses of COVID-19 outcomes and other diseases at the identified sQTL loci**

| sQTL locus (rsID) | chr | pos (b38) | EA | NEA | Intron junction | COVID-19 outcome | PP* between sQTL and COVID-19 | Other outcome | PP* between COVID-19 and other outcome |
|---|---|---|---|---|---|---|---|---|---|
| ATP11A (rs12585036) | 13 | 112881427 | C | T | chr13:112,875,941-112,880,546 ↑ (lung, WBC) | critical illness ↓ | 1.00 | idiopathic pulmonary fibrosis ↑ [PMID: 31710517] | 1.00 |
| DPP9 (rs12610495) | 19 | 4717660 | A | G | chr19:4,714,337-4,717,615 ↑ (lung) | critical illness ↓ | 0.96 | idiopathic pulmonary fibrosis ↓ [PMID: 31710517] | 1.00 |
| NPNT (rs34712979) | 4 | 105897896 | G | A | chr4:105,898,001-105,927,336 ↑ (lung) | critical illness ↑ | 1.00 | FEV1/FVC ratio↑ [PMID: 30804560] | 1.00 |
| | | | | | | | | COPD↓ [PMID: 30804561] | - |
| | | | | | | | | Asthma↓ [PMID: 31959851] | - |
| OAS1 (rs10774671) | 12 | 112919388 | G | A | chr12:112,917,700-112,919,389 ↑ (lung) | critical illness ↓ | 0.99 | systemic lupus erythematosus ↓ [PMID: 33272962] | - |
| | | | | | | | | chronic lymphocytic leukemia↑ [PMID: 28165464] | - |
| MUC1 (rs4072037) | 1 | 155192276 | C | T | chr1:155,192,310:155,192,786 ↑ (lung) | reported infection ↑ | 0.97 | inflammatory bowel disease ↓ [PMID: 28067908] | 1.00 |
| | | | | | | | | gastric cancer ↓ [PMID: 26098866] | - |
| | | | | | | | | gout ↑ [PMID: 33959723] | - |
| | | | | | | | | Urate level ↑ [PMID:33462484] | 1.00 |
| PMF1 (rs1052067) | 1 | 156236330 | A | G | chr1:156,233,728:156,236,349↑ (lung, whole blood) | reported infection ↓ | 0.95 | Testicular germ cell tumor ↑ [PMID: 28604728] | - |
| | | | | | | | | Serum creatinine levels ↓ [PMID: 34594039] | - |
| | | | | | | | | Intracerebral hemorrhage ↓ [PMID: 24656865] | - |
| | | | | | | | | Ischemic stroke ↓ [PMID: 29531354] | 0.30 |

aPP: a posterior probability that there is an association for two outcomes in GWASs, which is driven by the same causal variant. PP was only estimated when there is an available GWAS summary statistics of European ancestry. All significant results listed in Table 1 were estimated by Wald ratio.
*EA effect allele, NEA non-effect allele.*

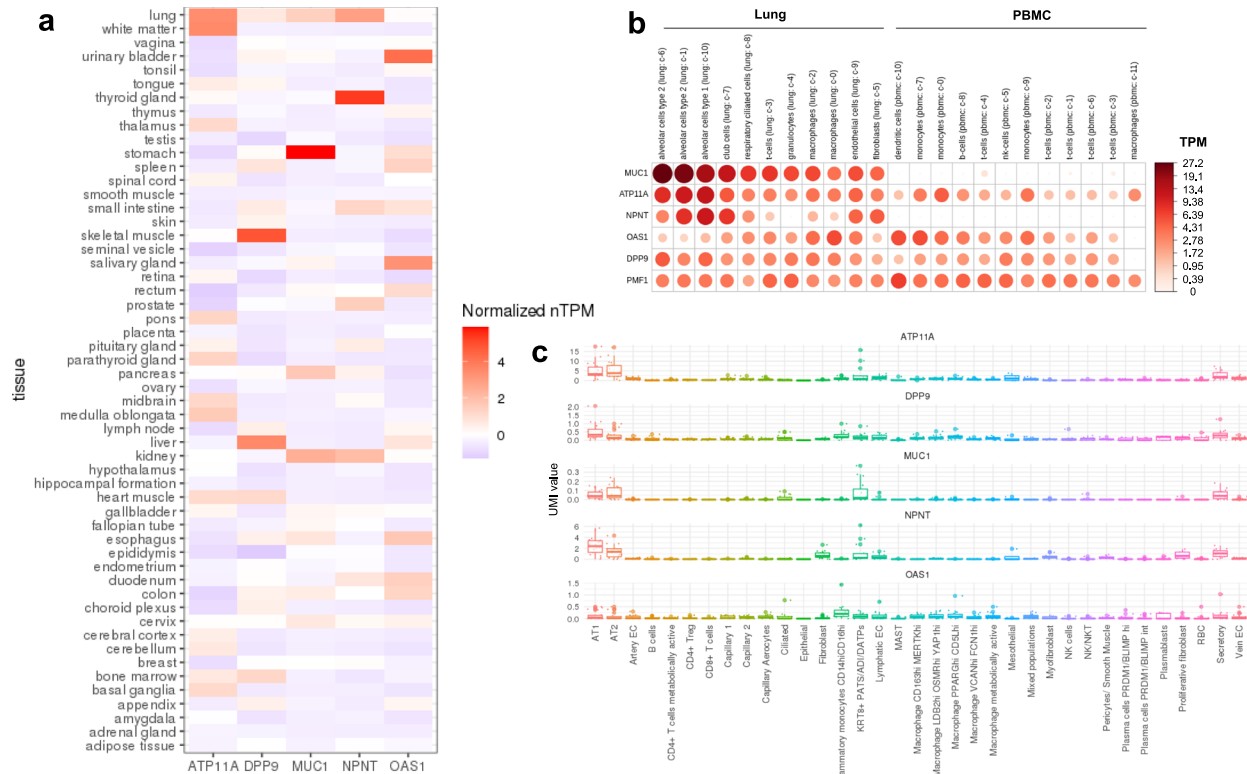

**Fig. 4 | Gene expression in lung and peripheral blood mononuclear cells (PBMCs). a** The consensus transcript expression levels summarized per gene in 54 tissues in Human Protein Atlas (HPA), which was calculated as the maximum transcripts per million value (TPM) value for each gene in all sub-tissues categories of each tissue, based on transcriptomics data from HPA and GTEx. The consensus transcript expression level for *PMF1* was not available in HPA. **b** RNA single cell type tissue cluster data (transcript expression levels summarized per gene and cluster) of lung (GSE130148) and peripheral blood mononuclear cell (PBMC) (GSE112845) were visualized using $log_{10}$(protein-transcripts per million [pTPM]) values. Each c-X annotation is taken from the clustering results performed in HPA. **c** Single-cell RNA expression profile of 23 lung COVID-19 autopsy donor tissue samples from GSE171668[20]. The RNA expression of *PMF1* was not detected in this dataset. The mean value of RNA expression of the cells annotated in the same subcategory was represented as a dot per each sample. The cell type annotation was manually performed in the original publication. AT1 alveolar type 1 epithelial cells, AT2 alveolar type 2 epithelial cells, EC endothelial cells, *KRT8*+ PATS/ADI/DATPs *KRT8*+ pre-alveolar type 1 transitional cell state, MAST mast cells, XXhi XX (gene expression) ^high cells, RBC red blood cells. Lower edge of the whisker: the lowest value within 1.5 * IQR of the hinge, lower hinge: 25% quantile, horizontal line contained within the box: median value, upper hinge: 75% quantile, the upper edge of the whisker: the highest value that is within 1.5 * IQR of the hinge.

severity. Interestingly, we found the opposite direction of effects between IPF and COVID-19 in the sQTL for *ATP11A*, where the rs12585036-C allele was associated with increasing risk of IPF and decreased risk for COVID-19 severity. The *NPNT* sQTL had similar trend for the rs34712979-A allele, which was protective for COVID-19 severity, but was associated with increased risk of COPD[24] (i.e. lower FEV1/FVC ratio, a spirometry measurement used to diagnose COPD)[33,34] and with increased risk of asthma[35]. The *OAS1* sQTL (rs10774671:A > G, protective allele for COVID-19 severity) was associated with a decreased risk of systemic lupus erythematosus (in the East Asian population)[36] and with an increased risk of chronic lymphocytic leukemia[37]. The *MUC1* sQTL (rs4072037:T > C, risk allele for SARS-CoV-2 infection) was associated with decreased risk of gastric cancer[38], gout[39], and inflammatory bowel disease[40]. The *PMF1* sQTL (rs1052067:G > A, protective allele for SARS-CoV-2 infection) was associated with increased risk of testicular germ cell tumor[41], decreased risk intracerebral haemorrhage and ischemic stroke[42] and decreased levels of serum creatinine[43].

We performed colocalization analyses in each sQTL locus between COVID-19 outcomes and the other associated diseases/phenotypes if the effects were supported in the European ancestry population and there was GWAS summary data available. IPF[32] and critical illness due to COVID-19 were highly colocalized at the *ATP11A* and *DPP9* sQTL loci with a posterior probability of 1.00. The FEV1/FVC ratio[33] also colocalized with critical illness due to COVID-19 with a

posterior probability of 1.00 at the *NPNT* locus. Inflammatory bowel disease[40] and urate level (which is a cause of gout) also colocalized with reported SARS-CoV-2 infections at the *MUC1* locus (both posterior probabilities are 1.00). We found no GWAS summary statistics available for the other diseases (Table 1). These lines of evidence suggest that IPF and COVID-19 severity, COPD/asthma and COVID-19 severity, and IBD/gout and COVID-19 susceptibility may share causal genetic determinants at these loci, respectively. The full results and the data used are summarized in Table 1.

**The tissue and cell-type specific expression of the associated genes**

To assess the relevant tissues and cell-types for the genes whose transcriptional splicing was identified by MR, we evaluated the transcriptional expression in lung and peripheral blood mononuclear cell (PBMC) of healthy controls, as well as in lung of COVID-19 patients. In the consensus transcript expression levels from Human Protein Atlas (HPA)[44] and GTEx, the expression of *ATP11A*, and *NPNT* were highly enriched in normal lung tissue (Fig. 4a). At a single-cell resolution, in normal lung, *MUC1*, *ATP11A* and *NPNT* were specifically enriched in alveolar type 1 and 2 epithelial cells (Fig. 4b), which are known to play important roles in regeneration of alveolar epithelium following lung injury[45]. In 23 lung COVID-19 autopsy donor tissue samples from GSE171668[20], the expression of *ATP11A, DPP9, MUC1*, and *NPNT* were also enriched in alveolar type 1 and 2 epithelial cells (Fig. 4c). Taken

together, these expression evidence suggests that the transcriptional splicing−COVID-19 outcome relationships for *ATP11A, DPP9, MUC1,* and *NPNT* were likely to be relevant in lung tissue and these signals may be especially important in alveolar epithelial cells.

## Discussion

Despite current vaccines and therapeutic options, hospitalization for COVID-19 remains high in many countries. Thus, there remains a need for additional therapies, which in turn, requires target identification and validation[46]. In this large-scale two-sample MR study of alternative splicing assessed for their effect upon three COVID-19 outcomes in up to 122,616 COVID-19 cases and 2,475,240 population controls, we provide evidence that alternative splicing of *OAS1, ATP11A, DPP9* and *NPNT* in lung influences COVID-19 severity, and alternative splicing of *MUC1* and *PMF1* in lung influences COVID-19 susceptibility. Moreover, these genetic mechanisms are likely shared with other diseases, such as *ATP11A* and *DPP9* with IPF.

There are multiple sources of biological evidence which might support the relevance of *ATP11A* in COVID-19. *ATP11A* is a member of the P4-ATPase family, which codes for a phospholipid flippase at the plasma membrane and translocates phosphatidylserine from the outer to the inner leaflet of plasma membranes. Cells that undergo apoptosis, necroptosis or pyroptosis expose phosphatidylserine on their surface through P4-ATPase, of which *ATP11A* is cleaved by a caspase during apoptosis[47]. While less functionally defined, given the high expression of *ATP11A* in the alveolar type 2 epithelial cells, the *ATP11A* sQTL (rs12585036) may impact the cell death of alveolar type 2 epithelial cells, the resident stem cell population in lung[48], and may impair regeneration of alveolar epithelium following lung injury. However, these hypotheses require further study.

Our MR analysis showed that the increased excision of intron junction chr19:4,814,337−4,717,615 in the *DPP9* gene, which is preferentially spliced in ENST00000599248, was associated with reduced risk of COVID-19 severity. Although it is not fully understood how the *DPP9* sQTL (rs12610495) regulates alternative splicing of *DPP9*, the rs12610495-G allele creates a GGGG motif (from GGAG sequence), which may have an effect on alternative splicing of some genes[49]. DPP9 interacts with NLRP1 and represses inflammasome activation[50] and pyroptosis, which are now recognized as important mechanisms interrupting the viral replication cycle and preventing viral amplification of SARS-CoV-2[51]. Moreover, targeting inflammasome-mediated hyperinflammation in COVID-19 patients may also prevent chronic phase of COVID-19 pathophysiology in vivo[51].

Our MR analyses found that *MUC1* splicing was associated with susceptibility of SARS-CoV-2 infection. *MUC1* is a mucin, which is also called KL−6 in humans, and the serum level of KL-6 is used as a biomarker for some interstitial lung diseases[52]. rs4072037-T allele was associated with decreased levels of the intron junction of chr1:155,192,310−155,192,786, which corresponds to transcripts with an alternative 27 bp intron retention event at the start of exon 2[28]. *MUC1* exon 2 harbors a variable number of tandem repeats (VNTR) that contains 20 to 125 repeats of a 60 bp coding sequence that determines the length of a heavily glycosylated extracellular domain[53]. Although rs4072037 may or may not control the alternative splicing of VNTR region[38], in addition to the alternative 27 bp intron retention[28], the VNTR length of *MUC1* is associated with several renal phenotypes in a recent study in UK Biobank[54]. The *MUC1* sQTL was also associated with IBD with high colocalization evidence. IBD is a disease with disrupted intestinal epithelial barrier and is suggested to be associated with gut dysbiosis[55]. Taken together, it is plausible that alternative splicing of *MUC1* in lung alveolar cells has an impact on SARS-CoV-2 infection.

This study has limitations. First, the excised intron junction was quantified in an annotation-free manner using LeafCutter[21], without respect to the level of transcripts nor isoforms. It is thus important for future work to map those disease-relevant alternative splicing events to the corresponding isoform or protein product, by means of emerging technologies such as long-read sequencing[56] and high-throughput protein quantification[57]. We anticipate our findings, and others, will motivate the ongoing effort to do so. Second, we used MR to test the effect of alternative splicing measured in a non-infected state since the effect of the *cis*-sQTLs upon alternative splicing was estimated in individuals who had not been exposed to SARS-CoV-2. Given the dynamic gene regulation of splicing during infection[58], alternative splicing could be changed once a person experiences SARS-CoV-2 infection. Thus, the MR results presented in this paper should be interpreted as an estimation of the effect of alternative splicing during uninfected state. Future studies may help to clarify if the same *cis*-sQTLs regulate alternative splicing during infection. Third, it was not our goal to identify all alternative splicing that affects COVID-19 outcomes, but rather to provide strong evidence for a small set of genes with strong MR and colocalization evidence. Thus, we acknowledge a high false-negative rate of our study design. Fourth, as we used the data from individuals of European descent, we could not confirm that our findings could be transferrable to individuals of other populations. Fifth, while our findings demonstrate the effects of alternative splicing in lung and whole blood since these are relevant to COVID-19 severity, we recognize that these splicing events may not be unique to these tissues and thus the estimated effects of splicing may represent the action of these same alternative transcripts in other tissues. Lastly, a recent paper[59] reported that the shared genetic signal between IPF and COVID-19 outcomes at the *DPP9* and *ATP11A* loci are likely driven by the difference of total expression in whole blood, which was supported by colocalization using eQTLGen[60] dataset. However, eQTLGen consists of multi-ancestry cohorts and could be affected by the bias due to LD. Moreover, we demonstrated that both *DPP9* and *ATP11A* expression were enriched in alveolar cells in lung, compared to whole blood. These findings support the hypothesis that alternative splicing of *DPP9* and *ATP11A* in lung is important both for IPF and COVID-19 severity.

Our results provide rationale to explore the targeting of alternative splicing in lung as a treatment for respiratory diseases[61,62] by means of therapeutic modalities such as splice-switching oligonucleotides (SSOs). SSOs are a type of antisense oligonucleotides which are generally 15−30 nucleotides in length and are designed as complementary to specific regions of mRNA with increased stability against endogenous nucleases due to chemical modifications[63]. SSOs can prevent splicing promoting factors from binding to the target pre-mRNA, which can modulate alternative splicing[64]. Some SSOs have been already approved by FDA, such as eteplirsen for Duchenne's muscular dystrophy, which induces the skipping of exon 51 of the *DMD* gene, and nusinersen for spinal muscle atrophy, which induces the expression of exon 7 of the *SMN2* gene[63]. Although delivery to target tissues after systemic administration has been a key challenge in the development of SSO drugs[63,65], lung might be advantageous tissue to target since it is possible to deliver drugs directly to lung through inhalation.

In conclusion, we have used genetic determinants of alternative splicing and COVID-19 outcomes obtained from large-scale studies and found compelling evidence that splicing events in *OAS1, ATP11A, DPP9, NPNT, MUC1,* and *PMF1* have causal effects on COVID-19 severity and susceptibility. Interestingly, the available evidence suggests shared genetic mechanisms for COVID-19 severity with IPF at the *ATP11A* and *DPP9* loci, and with chronic obstructive lung diseases at the *NPNT* locus. Taken together, our study highlights the importance of alternative splicing both in COVID-19 and other diseases, which could be further investigated for drug discovery programs.

## Methods

### Splicing quantitative trait loci (sQTL) GWASs

We obtained all conditionally independent sQTLs in lung and whole blood that act in *cis* (in a +/−1 Mb window around the transcription

start site of each gene) with normalized intron excision ratios from LeafCutter[21] (5% FDR per tissue) in GTEx v8[17]. Conditionally independent sQTLs were mapped using stepwise regression in GTEx consortium[17]. Intron excision ratios are the proportion of reads supporting each alternatively excised intron identified by LeafCutter[21]. We selected lung and whole blood since they are the two major tissues relevant to the pathophysiology of acute SARS-CoV-2 infection[20]. We obtained the effect estimate of each conditionally independent *cis*-sQTL from the data mapped in European-American subjects ($N = 452$ for lung and $N = 570$ for whole blood) in GTEx v.8[17]. If the *cis*-sQTL was missing in the summary data mapped in European-ancestry subjects, we removed those *cis*-sQTLs in the analyses.

For the key sQTLs, we created violin plots to visualize the normalized intron excision ratios stratified by sQTL genotypes. Normalized intron excision ratio was obtained from GTEx publicly available sQTL phenotype matrices (https://www.gtexportal.org/home/datasets). The individual-level genotypes were obtained from imputed genotype data in GTEx for which we achieved access through dbGaP. The samples with available data of both normalized intron excision ratio and sQTL genotypes were used to create violin plots ($N = 232 - 290$). We also drew sashimi plots, diagrams which combine the information of read coverage along a gene with curves connecting splice sites supported by RNA-seq data. We obtained individual RNA-seq mapped bam files for 514 lung samples in GTEx v.8 through dbGaP. We used ggsashimi (https://github.com/guigolab/ggsashimi) R package v1.1.5 to visualize splice junction usage per sQTL genotype. The mean number of reads supporting the splicing events per each genotype group are shown in the sashimi plots, which were adjusted for the average expression (counts per million: CPM) of the region including the cluster to which the index intronic junction belongs and the exons at both ends. CPM was calculated by (the mapped read counts of the region / the total read counts) $\times 10^{-6}$ (Supplementary Data 9).

## COVID-19 GWASs

To assess the association of *cis*-sQTLs with COVID-19 outcomes, we used COVID-19 GWAS meta-analyses results of European-ancestry subjects from the COVID-19 Host Genetics Initiative (HGI) release 7[10] (https://www.covid19hg.org/results/r7/). The outcomes tested were critical illness, hospitalization, and reported SARS-CoV-2 infection (named A2, B2, and C2, respectively by the COVID-19 HGI).

Critically ill COVID-19 cases were defined as those individuals who were hospitalized with laboratory-confirmed SARS-CoV-2 infection and who required respiratory support (invasive ventilation, continuous positive airway pressure, Bilevel Positive Airway Pressure, or continuous external negative pressure, high-flow nasal or face-mask oxygen) or who died due to the disease. Simple supplementary oxygen (e.g. 2 l/min via nasal cannula) did not qualify for case status. Hospitalized COVID-19 cases were defined as individuals hospitalized with laboratory-confirmed SARS-CoV-2 infection (using the same microbiology methods as for the critically ill phenotype), where hospitalization was due to COVID-19 related symptoms. Reported SARS-CoV-2 infection was defined as laboratory-confirmed SARS-CoV-2 infection or electronic health record, ICD coding or clinically confirmed COVID-19, or self-reported COVID-19 (for example, by questionnaire), with or without symptoms of any severity. Controls were defined in the same way across all three outcomes above as everybody that is not a case—for example, population controls.

In a sensitivity analysis, we also used another hospitalization phenotype (named B1 in the COVID-19 HGI), where cases were hospitalized COVID-19 cases and controls were defined as non-hospitalized individuals with laboratory-confirmed SARS-CoV-2 infection.

## Two-sample Mendelian randomization

We used two-sample mendelian randomization (MR) analyses using "TwoSampleMR v0.5.6" R package[66] to screen and test the potential role of alternative splicing in lung and whole blood to influence COVID-19 outcomes. In two-sample MR, the effect of genetic variants on the exposure and outcome are taken from separate GWASs. Two-sample MR often improves statistical power compared to single-sample MR, where sample sizes are smaller. MR is less affected by confounding and reverse causality than observational epidemiology studies since genotypes are essentially randomly assigned at conception and not influenced by the disease outcome. The MR framework is based on three main assumptions: First, the genetic variants are robustly associated with the exposure (i.e. a lack of weak instrument bias). We validated that all *cis*-sQTLs had a F-statistic >10, corresponding to T-statistics >3.16, which indicate a low risk of weak instrument bias in MR analyses[67]. Second, the genetic variants are not associated with any confounding factors for the relationship between the exposure and the outcome. Third, the genetic variants have no effect on the outcome that is independent of the exposure (i.e. a lack of horizontal pleiotropy), which is the most challenging assumption to assess. Nevertheless, in order to reduce the risk of horizontal pleiotropy, we selected *cis*-sQTLs as instrumental variables, as *cis*-genetic variants that reside close to the genes are more likely to have an effect on the outcomes by directly influencing the alternative splicing, thus reducing potential horizontal pleiotropy. Palindromic *cis*-sQTLs with minor allele frequencies (MAF) > 0.42 were removed prior to MR to prevent allele-mismatches. We also removed genetic variants within MHC region to reduce the risk of bias from LD. For alternative splicing with a single (sentinel) *cis*-sQTL, we used Wald ratio to estimate the effect of each splicing event on each of the three COVID-19 outcomes. For any alternative splicing event with multiple conditionally independent *cis*-sQTLs, an inverse variance weighted (IVW) method was used to meta-analyze their combined effects. After harmonizing the *cis*-sQTLs with COVID-19 GWASs, a total of 5724 splicing events in 4329 genes (5807 matched *cis*-sQTLs) in lung and a total of 3658 splicing events in 2671 genes (3658 matched *cis*-sQTLs) in whole blood were used for the MR analyses across the three COVID-19 outcomes. We applied the Bonferroni corrected p-value ($5.1 \times 10^{-6}$) to adjust for the number of tests performed ($N = 27,230$, 5546 tests between alternative splicing in lung and critical illness, 5570 tests between alternative splicing in lung and hospitalization, 5686 tests between alternative splicing in lung and reported SARS-CoV-2 infection, 3453 tests between alternative splicing in whole blood and critical illness, 3451 tests between alternative splicing in whole blood and hospitalization, 3524 tests between alternative splicing in lung and reported SARS-CoV-2 infection).

## Colocalization analyses

Next, we evaluated whether the splicing events and COVID-19 outcomes shared a common etiological genetic signal and that the MR results were not biased by linkage disequilibrium (LD) using colocalization analyses. Specifically, for each of these MR significant splicing events, a stringent Bayesian analysis was implemented in "coloc v5.1.0.1" R package with default settings to analyze all variants in 1MB genomic locus centered on the *cis*-sQTL. Colocalizations with posterior probability for high colocalization (PP > 0.8), that is, that there is an association for both splicing events and COVID-19 outcomes, and they are driven by the same causal variant were considered to colocalize, which means that the exposure and the outcome shared a single causal variant.

## Mendelian randomization and colocalization with eQTL GWASs from GTEx

To understand if the total gene expression levels in lung and whole blood were associated with COVID-19 outcomes, without respect to the splicing or isoforms, we similarly performed MR and colocalization analyses using expression quantitative trait loci (eQTLs) in lung and whole blood from GTEx v.8[17] (lung: $N = 452$ of European ancestry, and whole blood: $N = 570$ of European ancestry) by restricting the regions

within 1 Mb of each QTL. The genetic instruments were conditionally independent eQTLs for the prioritized sQTL genes in lung and/or whole blood, all of which had strong support for colocalization between sQTLs and COVID-19 outcomes.

### Influence of identified sQTLs on other diseases

We assessed the effects of the sQTLs which influence COVID-19 outcomes on other diseases. Pleiotropic search was performed using Open Targets Genetics[30,31] (https://genetics.opentargets.org) and identified any GWAS lead variants that are in LD ($r^2 > 0.80$) with those sQTLs. Colocalization analyses were performed using "coloc v5.1.0.1" R package when there were available GWAS summary statistics of European ancestry.

### The tissue and cell-type specific expression of the associated genes

To assess the relevant tissues and cell-types for the genes whose transcriptional splicing were identified by MR, we evaluated the transcriptional expression in lung and peripheral blood mononuclear cell (PBMC) of healthy controls, as well as in lung of COVID-19 patients. We first downloaded consensus transcript expression levels summarized per gene in 54 tissues in Human Protein Atlas (HPA)[44], which was calculated as the maximum transcripts per million value (TPM) value for each gene in all sub-tissues categories of each tissue, based on transcriptomics data from HPA and GTEx. We also downloaded the single-cell type transcriptomic analyses, where we used all cell types in lung (originally GSE130148[68]) and peripheral blood mononuclear cell (PBMC) (originally GSE112845[69]). We visualized RNA single cell type tissue cluster data (transcript expression levels summarized per gene and cluster), using $\log_{10}$(protein-transcripts per million [pTPM]) values with "corrplot v0.92" R package. Lastly, we obtained single-cell RNA expression profile of 23 lung COVID-19 autopsy donor tissue samples from GSE171668[20]. We calculated "pseudo-bulk" RNA expression per each cell type by taking the mean value of RNA expression of the cells annotated in the same subcategory.

### Inclusion and ethics statement

All collaborators of this study have fulfilled the criteria for authorship required by Nature Portfolio journals have been included as authors, as their participation was essential for the design and implementation of the study. This research was not severely restricted or prohibited in the setting of the researchers. GTEx protected access data was applied through dbGaP upon ethical approval from Jewish General Hospital (ethics number 2024-3794). This research does not result in stigmatization, incrimination, discrimination or personal risk to participants.

### Reporting summary

Further information on research design is available in the Nature Portfolio Reporting Summary linked to this article.

## Data availability

Summary statistics for eQTLs and sQTLs from GTEx v8[17] are available in the GTEx website (gtexportal.org/home/datasets). The GTEx protected access data are available under restricted access, access can be obtained through dbGaP (application ID: 32756). Summary statistics for the COVID-19 outcomes release 7 are available in the COVID-19 Host Genetics Initiative website[70] (https://www.covid19hg.org/results/r7/). The consensus transcript expression levels and RNA single cell type tissue cluster data are available at the Human Protein Atlas website (www.proteinatlas.org/about/download). RNA single cell type tissue cluster data for lung and PBMC was originally obtained from GSE130148[68] and GSE112845[69], respectively. The processed single-cell RNA expression profile of 23 lung COVID-19 autopsy donor tissue sample are available at the Gene Expression Omnibus (GEO, https://www.ncbi.nlm.nih.gov/geo/) under accession code GSE171668[20].

## Code availability

All code for data management and analysis is archived online at github.com/richardslab/COVID19-sQTLMR for review and reuse[71]. We used ggsashimi (https://github.com/guigolab/ggsashimi) R package v1.1.5 to visualize splice junction usage per sQTL genotype. We used TwoSampleMR v0.5.6 R package[66] to run two-sample mendelian randomization (MR) analyses. Colocalization analyses were performed using coloc v5.1.0.1 R package. We visualized RNA single cell type tissue cluster data using corrplot v0.92 R package.

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

## Acknowledgements

We would like to thank Dr. Luis Barreiro for his expertise and assistance in writing the manuscript. The Richards research group is supported by the Canadian Institutes of Health Research (CIHR: 365825; 409511, 100558, 169303), the McGill Interdisciplinary Initiative in Infection and Immunity (MI4), the Lady Davis Institute of the Jewish General Hospital, the Jewish General Hospital Foundation, the Canadian Foundation for Innovation, the NIH Foundation, Cancer Research UK, Genome Québec, the Public Health Agency of Canada, McGill University, Cancer Research UK [grant umber C18281/A29019] and the Fonds de Recherche Québec Santé (FRQS). J.B.R. is supported by a FRQS Mérite Clinical Research Scholarship. Support from Calcul Québec and Compute Canada is acknowledged. TwinsUK is funded by the Welcome Trust, Medical Research Council, European Union, the National Institute for Health Research (NIHR)-funded BioResource, Clinical Research Facility and Biomedical Research Centre based at Guy's and St Thomas' NHS Foundation Trust in partnership with King's College London. These funding agencies had no role in the design, implementation or interpretation of this study. T.N. is supported by a research fellowship of the Japan Society for the Promotion of Science for Young Scientists (22KJ1190, 22J30004) and this work was supported by Grant-in-Aid for Scientific Research(B) (JSPS KAKENHI Grant Number: 23H02917). J.W. is supported by the CIHR (476575). B.G.G. is supported by Wellcome Trust grant 221680/Z/20/Z.

## Author contributions

Conception and design: T.N. and J.B.R. Formal analysis: T.N. and J.W. Data curation: T.N., J.W., and Y.F. Interpretation of data: T.N., J.W., Y.F., R.J.A., B.G.G., and J.B.R. Funding acquisition: T.N., J.B.R. Investigation: T.N., J.W., and Y.F. Methodology: T.N., Y.F., and J.B.R. Project administration: T.N., D.A., J.B.R. Resources: T.N., J.B.R. Supervision: T.N., Y.F., and J.B.R. Validation: T.N., J.W., Y.F., and J.B.R. Visualization: T.N. and J.W. Writing—original draft: T.N. Writing—review and editing: T.N., J.W., Y.F., R.J.A., B.G.G., D.A., S.Z., and J.B.R. All authors were involved in further drafts of the manuscript and revised it critically for content.

All authors gave final approval of the version to be published. The corresponding authors attest that all listed authors meet authorship criteria and that no others meeting the criteria have been omitted.

## Competing interests

T.N. has received speaking fee from Boehringer Ingelheim for the talks unrelated to this research. J.B.R.'s institution has received investigator-initiated grant funding from Eli Lilly, GlaxoSmithKline and Biogen for projects unrelated to this research. J.B.R. is the CEO of 5 Prime Sciences (www.5primesciences.com), which provides research services for biotech, pharma and venture capital companies for projects unrelated to this research. Y.F. is an employee of 5 Prime Sciences. All other authors declare no competing interests.
