## [Peer Review File · Nature Communications]

Alternative splicing in lung influences COVID-19 severity and respiratory diseasesREVIEWER COMMENTS

Reviewer #1 (Remarks to the Author):

See attached document

Overall summary

This manuscript uses Mendelian Randomisation (MR) to examine whether alternatively spliced RNA transcripts play a causal role in COVID-19 severity. As such, the paper uses existing publicly available data (splice QTL from GTEx and COVID-19 GWAS summary statistics from COVID19 HGI). There is no new primary data generation. The MR analysis approach was carefully considered and generally very sound with one exception (see below). Appropriate sensitivity analyses and checks (e.g. colocalization) were performed. The manuscript was very clearly written and the Figures were generally clear. To my knowledge this is the first transcriptome-wide assessment of the causal role of spliced isoforms in an MR framework. The authors identify a number of examples where the statistical genetic evidence for a causal role in COVID-19 outcomes (and some other diseases) is strong. These are potentially biologically interesting, and this is expanded on clearly in the Discussion. No additional validation experiments/de novo data generation are performed, and the precise mechanism underlying the putative causal associations remains unclear. However, I do not think it is reasonable to expect this for an MR paper in Nature Communications. Hopefully the results here will motivate further such studies. A 'high-level' criticism is that the authors are not able to link the alternatively spliced transcripts to specific changes in protein structure or function. Understanding this will be a key starting point for insight into the biological mechanism underpinning the associations observed. The authors speculate that their findings may lead to new therapies for COVID-19; I think this may be a stretch and I suspect the translational impact of the findings will be limited, but this does not detract from the value of the work in terms of investigation of pathogenesis of the host response to COVID-19.

Overall, I was very positive about the manuscript and would support publication of a revised manuscript.

Major point

I have one major methodological concern which is entirely addressable (and indeed the authors have already performed the necessary analysis as part of their sensitivity analysis). This concern relates to the issue of analysing distinct ancestry populations together which is problematic for two reasons (as I'm sure the authors are well aware): confounding of genetics associations by population stratification and distinct LD patterns in different ancestry groups. The latter can impact various steps in the analysis: it may affect imputation, it can affect the p-values across a locus (as p-values are affected by MAF), and it can affect colocalization results. These issues are particularly pertinent in the context of a two-sample MR study where the genetic effect on the exposure is estimated in a different dataset to the genetic effect on the outcome.

Specifically, in the primary analysis the authors use splice QTLs from GTEx as genetic instruments, where for lung 452 of 515 individuals (~88%) were European ancestry, and 570/670 (~85%) for blood. A similar large majority of the HGI COVID-19 GWAS data was from European-ancestry individuals. The ancestry of the non-European ancestry individuals in GTEx and HGI are not listed in this manuscript and it is not made clear whether they are well-matched in one dataset versus the other; determining this would require going back to the

source datasets. Regardless of this, the combined ancestry analysis presents methodological problems.

The authors justify their initial use of sQTL data and COVID GWAS data from all individuals on the basis of not wishing to ‘discard data from minority populations’, and cite a commentary by Ben-Eghan et al (ref 21) to support this approach. Unfortunately, the commentary by Ben-Eghan is brief and superficial: it does not consider differing LD patterns at all; it provides no empirical or simulation data to demonstrate a robust approach for negating the problems of population stratification; and it falsely characterises attempts to avoid population stratification as analyst laziness, when in fact the genomics community has developed robust methodology to minimise false positives over the past two decades. There is no doubt that there is a pressing need for an expansion of genomic studies in non-European ancestry populations, and I am hugely supportive and excited by such endeavours. However, the issue of the European-centric bias can only be tackled by primary data generation at scale. Performing non-robust analyses is not the solution.

The authors are clearly aware of all these issues, as they go on to perform a European-ancestry only sensitivity analysis, and they perform their colocalization analysis only in the European-ancestry subjects. It was not entirely clear to me how many (if any) genes dropped out following the European-ancestry only analysis as the schematic in Figure 1 shows the results after both European-ancestry only analysis and colocalization testing. Given that European-ancestry individuals make up the great majority of the datasets, it may be that the results of the sensitivity analysis are very similar in this particular instance. However, my concern is that if published in its current form, the primary analysis provides justification for others to take a suboptimal approach where it may lead to false positive findings. Given their filtering strategy, the authors effectively discard the non-European data anyway (they deprioritize signals that are not found in the European-ancestry sensitivity analysis). Thus their current approach seems circuitous and indirect.

I would therefore recommend the authors present their European-ancestry only analysis as the primary analysis and provide the logic for this approach. Given the limited sample size of the non-European ancestry group, it is challenging to think of a way those data could be leveraged, particularly as this group may not be ancestrally homogeneous. If the authors are keen to utilise this data, perhaps they could check whether the effect sizes seem consistent with the European group even if the p-values are not “QTL” or GWAS level significant in the respective datasets. I note that some of the splice-QTLs that colocalise with COVID outcome signals vary considerably across ancestry groups in terms of allele frequency (e.g. rs12585036 at the ATP11A locus has MAF ~6% in 1kG AFR vs 21% in EUR). I would be open to any other innovative but robust ways the authors wish to use the non-European ancestry data in.

Minor points

-line 98-99: cis-sQTLs: the definition provided is circular: ‘cis-sQTLs act in cis’. Better to say they act on local genes.

-use of the term “SNPs” at various points. Do the authors really mean only SNPs, or are indels also included? In which case “genetic variants” would be more accurate.

-line 128-131: “we replicated the association of COVID-19 outcomes with OAS1 gene splicing...”. Can the authors clarify if this is really independent replication of this finding (i.e. in a different dataset) or is it simply re-finding this result in analysis based on the same underlying data (GTEx and HGI). Clearly the latter is much less compelling as “replication”.

-Can the authors clarify if any of the splice QTLs used as genetic instruments for the “final” set of significant genes are in high LD (r^2 0.8 or more) to any protein-coding variant? I.e. could the putative effect of a splice-affecting variant actually be due to a protein-coding one instead?

-line 252 “we evaluated the associated gene expression” ... there didn’t appear to be any association testing, rather just looking at tissue expression levels. Suggest rephrase to be simpler and clearer.

-The Methods said the Wald test was used for single variant MR, and IVW where there were multiple instruments. It would be helpful to have N variants used in a Word Table for the significant results (and the ids of those variants). Apologies if I have missed this – it was not apparent in the Supp Excel which just said “All – inverse variance weighted”.

-In the case where there are multiple independent variants, scatterplots of effect on exposure and effect on outcome would be a useful addition to check consistency.

-Fig 2: would be helpful to state if Wald test or IVW.

-Fig 3 & Ext data Fig 1: need to indicate whether sQTL and eQTL are whole blood or lung either on plot or legend.

-It would be helpful to visualise key splice QTLs with of plots of genotype vs isoform abundance (e.g. % of transcripts with whatever the feature of interest is – retained intron etc etc).

-Have the authors considered looking at whether of their splice QTLs with a significant MR causal effect are also protein QTLs in publically available datasets? Both blood and lung splicing could potentially read out as apparent or real altered protein abundance in plasma/serum (e.g. if an aptamer or antibody targets an epitope that is impacted by alternative splicing). This would add significantly to our understanding of the biological pathways.

-Did the authors check whether the splice QTLs used as instruments for the significant MR findings show any evidence of pleiotropy in terms of acting on other molecular traits? E.g. are any trans eQTLs?

-Supplementary Excel tables would benefit from more clarity in terms of column headers and explanations – perhaps best as accompanying legends in the manuscript file. E.g. it took me a while to understand “Method/SNP” in ST 1 – which I think shows the position of the instrument if a Wald test was used but says “All- inverse variance weighted” if multiple

instruments were used. I would suggest SNPs in one column with Method in another. Similarly, the exposure in ST1 is a paste of a genomic position and gene name. Is the genomic position the TSS, the intron position or the SNP used as the instrument?

Minor comments on the Discussion

-The authors identify shared genetic influences on IBD and COVID-19 at the MUC1 locus. It might be worth discussing how it is plausible that altered mucin could impact both COVID and IBD risk, given that IBD is a disease of epithelial surfaces and dysregulated microbiota appears to be involved in its pathogenesis. However, I accept this is speculative and is only a suggestion for the Discussion.

- The authors speculate that targeting gene splicing in COVID19 may be helpful therapeutically. This was interesting but I think unlikely to happen. There are much more tractable targets e.g. inflammatory molecules that are the target of existing drugs. The fact that even these anti-inflammatory therapies have not been taken forward speaks to the uphill battle for a new therapeutic in severe COVID (i.e. it has to demonstrate additional benefit in an RCT above and beyond steroids and IL6R inhibition).

-The authors present MR evidence of a causal effect as evidence of a good potential therapeutic target. I'm familiar with this logic and with the data suggesting drugs targeting molecules implicated by human genetics are more likely to be successful. However, I would suggest that MR's value in therapeutic target prioritization may be higher in chronic diseases (cardiovascular disease and LDL-cholesterol being the obvious MR exemplar). Genetics has proven less effective in identifying therapies in COVID19. Anti-IL6R works very well despite lack of any convincing COVID GWAS signal at the well-known common variant that affects IL6R cleavage. Conversely, human genetics implicates type 1 interferon genes and yet interferon therapy has not produced convincing results. I would suggest that in dynamic acute contexts like the host immune response to infection, going after the causal triggers may not work (by analogy, arresting the first rioter won't stop the riot once it's in full swing...).

-The suggestion of bronchoscopically delivered therapy also seems unrealistic. Bronchoscopies are generally avoided in COVID19 (aerosol generating) unless there is another indication such as a suspicion of alternative or concurrent infection or other diagnosis, and therefore this group is not one in whom SSOs would be appealing.

Reviewer #2 (Remarks to the Author):

This is an interesting paper in which the authors integrate GWAS data from the COVID-19 Host Genetics Initiative with splice QTL data from GTEx to identify genetic variants that influence COVID-19 susceptibility by modulating transcriptional splicing. While the results are interesting and the methods are technically sound, the manuscript would benefit from in depth editing to improve readability and clarify findings. In particular, more information is needed about the sQTLs highlighted as relevant to disease so that inferences can be made as to biological mechanism.

Major Comments

-This manuscript requires significant editing for grammar and English language

-The authors should provide information about the effect size and direction of effect for the sQTLs highlighted in the manuscript (OAS1, ATP11A, DPP9, NPNT, and MUC1). It would also be useful to include sashimi plots detailing the splicing events described in the text, and either include quantitative measurements of splice ratios or include box plots of splicing ratios by genotype so the reader can get a visual representation of the magnitude and direction of effect for each sQTL. There is currently insufficient data for the reader to understand what changes are occurring according to genotype and which changes are associated with disease. At the very least there should be a cartoon diagram describing the splice changes that are associated with genotype and disease.

-Leafcutter sQTL results typically include multiple introns grouped into clusters. In this data there only appears to be information for one intron per gene. What happened to the rest of the introns in the cluster? How was the best intron selected? Without information about the individual introns in the cluster it is difficult to characterize what is happening at the splice site.

-Lines 230-231 –The A allele of rs34712979 has also been shown to be associated with increased COPD risk (not just FEV1/FVC ratio) through an sQTL in lung tissue – please cite PMID 33173926

Minor Comments

-Line 128-131- please specify the variant ID and the direction of effect so that it is not necessary to refer to your previous paper to understand this result

-The authors should define a new term or abbreviation only once in the main text of the manuscript – for example sQTL is defined on lines 72, 81-82, 95, 364;

-Instead of the term “gene splicing” – “RNA” or “transcriptional” splicing would be more accurate

-lines 106-107: “A total of 4,477 genes in lungs ... contained conditionally independent cis-sQTLs that were also present in the GWAS meta-analyses...” This statement is unclear – do the authors mean that 4,477 genes in lung tissue, and 2,779 genes in whole blood contained splice sites that were associated with at least one SNP that was also tested in the GWAS? It would be helpful to clarify how many splice sites, genes, sQTL-SNPs and sQTL SNP-gene pairs are being referred to here (ideally in a table in the main text)

-lines 277-285 (description of splice-switching oligonucleotides) should be moved towards the end of the discussion, the current position does not flow logically.

We thank the Reviewers and Editors for their constructive feedback and the invitation to re-submit our work to *Nature Communications*. Below we address, point-by-point, the comments of the Reviewers. Our responses are in **blue font** and modified manuscript text are in **orange font** to improve readability and all changes to the manuscript have been denoted by line numbers.

REVIEWER COMMENTS

Reviewer #1 (Remarks to the Author):

Overall summary

This manuscript uses Mendelian Randomisation (MR) to examine whether alternatively spliced RNA transcripts play a causal role in COVID-19 severity. As such, the paper uses existing publicly available data (splice QTL from GTEx and COVID-19 GWAS summary statistics from COVID19 HGI). There is no new primary data generation. The MR analysis approach was carefully considered and generally very sound with one exception (see below). Appropriate sensitivity analyses and checks (e.g. colocalization) were performed. The manuscript was very clearly written and the Figures were generally clear. To my knowledge this is the first transcriptome-wide assessment of the causal role of spliced isoforms in an MR framework. The authors identify a number of examples where the statistical genetic evidence for a causal role in COVID-19 outcomes (and some other diseases) is strong. These are potentially biologically interesting, and this is expanded on clearly in the Discussion. No additional validation experiments/de novo data generation are performed, and the precise mechanism underlying the putative causal associations remains unclear. However, I do not think it is reasonable to expect this for an MR paper in Nature Communications. Hopefully

the results here will motivate further such studies. A 'high-level' criticism is that the authors are not able to link the alternatively spliced transcripts to specific changes in protein structure or function. Understanding this will be a key starting point for insight into the biological mechanism underpinning the associations observed. The authors speculate that their findings may lead to new therapies for COVID-19; I think this may be a stretch and I suspect the translational impact of the findings will be limited, but this does not detract from the value of the work in terms of investigation of pathogenesis of the host response to COVID-19.

Overall, I was very positive about the manuscript and would support publication of a revised manuscript.

We thank the Reviewer #1 for his/her supportive comments.

Major point

I have one major methodological concern which is entirely addressable (and indeed the authors have already performed the necessary analysis as part of their sensitivity analysis). This concern relates to the issue of analysing distinct ancestry populations together which is problematic for two reasons (as I'm sure the authors are well aware): confounding of genetics associations by population stratification and distinct LD patterns in different ancestry groups. The latter can impact various steps in the analysis: it may affect imputation, it can affect the p-values across a locus (as p-values are affected by MAF), and it can affect colocalization results. These issues are particularly pertinent in the context of a two-sample MR study where the genetic effect on the exposure is estimated in a different dataset to the genetic effect on the outcome.

Specifically, in the primary analysis the authors use splice QTLs from GTEX as genetic instruments, where for lung 452 of 515 individuals (~88%) were European ancestry, and 570/670 (~85%) for blood. A similar large majority of the HGI COVID-19 GWAS data was from European-ancestry individuals. The ancestry of the non-European ancestry individuals in GTEX and HGI are not listed in this manuscript and it is not made clear whether they are well-matched in one dataset versus the other; determining this would require going back to the source datasets. Regardless of this, the combined ancestry analysis presents methodological problems.

The authors justify their initial use of sQTL data and COVID GWAS data from all individuals on the basis of not wishing to 'discard data from minority populations', and cite a commentary by Ben-Eghan et al (ref 21) to support this approach. Unfortunately, the commentary by Ben-Eghan is brief and superficial: it does not consider differing LD patterns at all; it provides no

empirical or simulation data to demonstrate a robust approach for negating the problems of population stratification; and it falsely characterises attempts to avoid population stratification as analyst laziness, when in fact the genomics community has developed robust methodology to minimise false positives over the past two decades. There is no doubt that there is a pressing need for an expansion of genomic studies in non-European ancestry populations, and I am hugely supportive and excited by such endeavours. However, the issue of the European-centric bias can only be tackled by primary data generation at scale. Performing non-robust analyses is not the solution.

The authors are clearly aware of all these issues, as they go on to perform a European-ancestry only sensitivity analysis, and they perform their colocalization analysis only in the European-ancestry subjects. It was not entirely clear to me how many (if any) genes dropped out following the European-ancestry only analysis as the schematic in Figure 1 shows the results after both European-ancestry only analysis and colocalization testing. Given that European-ancestry individuals make up the great majority of the datasets, it may be that the results of the sensitivity analysis are very similar in this particular instance. However, my concern is that if published in its current form, the primary analysis provides justification for others to take a suboptimal approach where it may lead to false positive findings. Given their filtering strategy, the authors effectively discard the non-European data anyway (they deprioritize signals that are not found in the European-ancestry sensitivity analysis). Thus their current approach seems circuitous and indirect.

I would therefore recommend the authors present their European-ancestry only analysis as the primary analysis and provide the logic for this approach. Given the limited sample size of the non-European ancestry group, it is challenging to think of a way those data could be leveraged, particularly as this group may not be ancestrally homogeneous. If the authors are keen to utilise this data, perhaps they could check whether the effect sizes seem consistent with the European group even if the p-values are not “QTL” or GWAS level significant in the respective datasets. I note that some of the splice-QTLs that colocalise with COVID outcome signals vary considerably across ancestry groups in terms of allele frequency (e.g. rs12585036 at the ATP11A locus has MAF ~6% in 1kG AFR vs 21% in EUR). I would be open to any other innovative but robust ways the authors wish to use the non-European ancestry data in.

We sincerely appreciate the Reviewer #1’s thorough and convincing thoughts on the issue of analyzing distinct ancestry populations together (hereby called “joint-ancestry” analysis). In general, we agree with the Reviewer #1 and indeed we had already recognized the issue.

We have now decided to use the European-only analysis as the primary analysis and omit the joint-ancestry analysis, which essentially did not have clear advantages over the European-only analysis.

For interest and to directly answer the Reviewer #1's question, "It was not entirely clear to me how many (if any) genes dropped out following the European-ancestry only analysis", there were no genes that dropped out in European-ancestry only analysis, compared to "joint-ancestry analysis". We are now convinced by the Reviewer #1's opinion that "if published in its current form, the primary analysis provides justification for others to take a suboptimal approach where it may lead to false positive findings".

MR results from European-only analyses:

Our new primary analysis which focused on individuals of European ancestry identified another interesting alternative splicing which influences SARS-CoV-2 infection. We have now added the following sentences in the Results, page 7, lines 184-190:

“PMF1 sQTL, rs1052067:G>A, which increases the excision of the intron junction at chr1:156,233,728-156,236,349 by 1.4 SD in lung (Fig. 2F, Supplementary Fig. 1F) and by 1.5 SD in whole blood, was associated with reduced risk of SARS-CoV-2 infection (OR: 0.98, 0.98-0.99, $p=8.1 \times 10^{-7}$, both in lung, Fig. 3, Supplementary Table 1, and whole blood, Supplementary Table 2). rs1052067:G>A is a missense variant which replaces methionine with isoleucine at amino acid position 137 of the MANE transcript, ENST00000368277. This variant creates another transcript, ENST00000368279, with an alternative 61bp intron retention event at the start of exon 4 (Supplementary Fig. 1F).”

Minor points

-line 98-99: cis-sQTLs: the definition provided is circular: ‘cis-sQTLs act in cis’. Better to say they act on local genes.

We have now modified the sentence in the Results, on page 4, line 102:

Original: “We chose to examine only *cis*-sQTLs, which act *cis* to the coding genes”

Edit: “We chose to examine only *cis*-sQTLs, which are more likely to act on local coding genes”

-use of the term “SNPs” at various points. Do the authors really mean only SNPs, or are indels also included? In which case “genetic variants” would be more accurate.

We thank the Reviewer #1 for raising this point. As we have included indels as instrumental variables, we have now changed the term “SNP(s)” to “(genetic) variant(s)”, when the term includes indels.

-line 128-131: “we replicated the association of COVID-19 outcomes with OAS1 gene splicing...”. Can the authors clarify if this is really independent replication of this finding (i.e. in a different dataset) or is it simply re-finding this result in analysis based on the same underlying data (GTEx and HGI). Clearly the latter is much less compelling as “replication”.

This is an important point. While our first OAS1 manuscript¹ used the HGI data release 4 (including up to 14,134 COVID-19 cases and 1,284,876 controls of European ancestry), the current manuscript is based on the COVID-19 HGI release 7 (including up to 122,616 cases and 2,475,240 controls of European ancestry). For the critical

illness phenotype, the association become stronger with p-value of 6.4×10^{-10} in release 7 compared to 7.0×10^{-8} in release 4. Thus, although it is the re-analysis based on the same GTEx and a larger HGI dataset, we believe this serves as a partial replication using the additional ten-fold increase in cases when compared to release 4.

We have clarified the sentence by adding the following sentence in the Results, on page 5, lines 128-130:

“We first replicated the association of COVID-19 outcomes with OAS1 splicing, using an updated version of COVID-19 HGI release 7, which provides a 10-fold increase in case sample size.”

-Can the authors clarify if any of the splice QTLs used as genetic instruments for the “final” set of significant genes are in high LD ($r^2 > 0.8$ or more) to any protein-coding variant? I.e. could the putative effect of a splice-affecting variant actually be due to a protein-coding one instead?

We appreciate the Reviewer #1 for this insightful comment. We searched for any protein-coding variants that were in high LD ($r^2 > 0.8$) using LDlink² (high coverage 1000G WGS data) for the following six sQTL SNPs;

ATP11A rs12585036

DPP9 rs12610495

MUC1 rs4072037

NPNT rs34712979

OAS1 rs10774671

PMF1 rs1052067

We did identify some coding variant sQTLs and some sQTLs in LD with coding variants.

Specifically, rs1052067:G>A of *PMF1* is a missense variant which replaces methionine with isoleucine at amino acid position 137 of the MANE transcript, ENST00000368277. This variant creates another transcript ENST00000368279 with an alternative 61bp intron retention event at the start of exon 4. rs10774671 at *OAS1* had also n LD with; rs2660 ($r^2=0.97$) and rs1859330 ($r^2=0.87$). Given that rs10774671 is a well-known splice-acceptor variant which has been functionally validated³, it is more likely that rs10774671 is the causal variant for this splicing effect. None of the other variants had any proxy coding-variants.

We have added the following sentence in the Results, on page 5, lines 136-138:

“While rs10774671 is in LD with two coding variants, rs2660 ($r^2=0.97$) and rs1859330 ($r^2=0.87$), it is more likely that rs10774671 is the causal variant for the splicing effect, given that rs10774671 is a well-known splice-acceptor variant which has been functionally validated²⁴”

-line 252 “we evaluated the associated gene expression” ... there didn't appear to be any association testing, rather just looking at tissue expression levels. Suggest rephrase to be simpler and clearer.

We have now rephrased the sentence to be clearer in the Results, on page 11, lines 273-275:

Original: “To assess the relevant tissues and cell-types for the associated gene splicing – COVID-19 outcome relationships, we evaluated the associated gene expression in lung and peripheral blood mononuclear cell (PBMC) of healthy controls, as well as in lung of COVID-19 patients.”

Edit: “To assess the relevant tissues and cell-types for the genes whose transcriptional splicing was identified by MR, we evaluated the transcriptional expression in lung and peripheral blood mononuclear cell (PBMC) of healthy controls, as well as in lung of COVID-19 patients.”

-The Methods said the Wald test was used for single variant MR, and IVW where there were multiple instruments. It would be helpful to have N variants used in a Word Table for the significant results (and the ids of those variants). Apologies if I have missed this – it was not apparent in the Supp Excel which just said “All – inverse variance weighted”.

Table 1 shows all the significant MR results with rsids and we have now clarified by adding the following sentence in the legend.

“All significant results listed in Table 1 were estimated by Wald ratio.”

Supplementary Tables were also fixed as suggested, which is detailed in the answer to your question regarding “Supplementary Excel tables”.

sQTL locus (rsID)	chr	pos (b38)	EA	NEA	intron junction	COVID-19 outcome	Other outcome	PP*
ATP11A (rs12585036)	13	112881427	C	T	chr13:112,875,941-112,880,546 ↑ (lung, WBC)	critical illness ↓	idiopathic pulmonary fibrosis ↑ [PMID: 31710517]	1.00
DPP9 (rs12610495)	19	4717660	A	G	chr19:4,714,337-4,717,615 ↑ (lung)	critical illness ↓	idiopathic pulmonary fibrosis ↓ [PMID: 31710517]	1.00
NPNT (rs34712979)	4	105897896	A	G	chr4:105,898,001-105,927,336 ↑ (lung)	critical illness ↓	FEV1/FVC ratio ↓ [PMID: 30804560]	1.00
							COPD ↑ [PMID: 30804561]	-
							Asthma ↑ [PMID: 31959851]	-
OAS1 (rs10774671)	12	112919388	G	A	chr12:112,917,700-112,919,389 ↑ (lung)	critical illness ↓	systemic lupus erythematosus ↓ [PMID: 33272962]	-
							chronic lymphocytic leukemia ↑ [PMID: 28165464]	-
MUC1 (rs4072037)	1	155192276	C	T	chr1:155,192,310:155,192,786 ↑ (lung)	reported infection ↑	inflammatory bowel disease ↓ [PMID: 28067908]	1.00
							gastric cancer ↓ [PMID: 26098866]	-
							gout ↓ [PMID: 33832965]	-
							Urate level ↓ [PMID:33462484]	1.00

PMF1 (rs1052067)	1	156236330	A	G	chr1:156,233,728:156,236,349↑ (lung, whole blood)	reported infection ↓	Testicular germ cell tumor ↑ [PMID: 28604728]	-
							Serum creatinine levels ↓ [PMID: 34594039]	-
							Intracerebral hemorrhage ↓ [PMID: 24656865]	-
							Ischemic stroke ↓ [PMID: 29531354]	0.30

-In the case where there are multiple independent variants, scatterplots of effect on exposure and effect on outcome would be a useful addition to check consistency.

Although this is a relevant comment, all significant MR findings were driven by single variants. Therefore, this suggestion is not applicable.

-Fig 2: would be helpful to state if Wald test or IVW.

We have now stated that all results were estimated using Wald ratio in the legend as following.

“All significant results listed in Fig. 3. were estimated by Wald ratio.”

-Fig 3 & Ext data Fig 1: need to indicate whether sQTL and eQTL are whole blood or lung either on plot or legend.

We appreciate the Reviewer #1 for noting this. We have now stated in the title and the legend of Supplementary Fig. 2. (We have moved all LocusZoom plots to Supplementary Fig. 2.) that all eQTLs and sQTLs are in lung, as follows.

Title: “LocusZoom plots demonstrating the colocalization of the genetic determinants of mRNA expression, RNA splicing levels *in lung* and COVID-19 outcomes”

Legend: “LocusZoom plots of *e/sQTLs in lung, and COVID-19 outcomes within a 1MB region around each cis-sQTL in lung.*”

-It would be helpful to visualise key splice QTLs with of plots of genotype vs isoform abundance (e.g. % of transcripts with whatever the feature of interest is – retained intron etc etc).

We appreciate this highly relevant suggestion. We have now applied to dbGAP to secure access to the GTEx protected access data⁴ (individual-level genotype and RNA sequencing data). This access request was granted.

In Fig. 2., we have now visualized the effect of the QTLs on the normalized intron excision ratios using violin plots.

Fig. 2. The violin plots of normalized intron excision ratio stratified by the sQTL genotypes.

a) *OAS1*, b) *ATP11A*, c) *DPP9* and d) *NPNT* show the sQTLs in dark orange that are associated with COVID-19 severity. e) *MUC1* and f) *PMF1* show the sQTLs in dark blue that are associated with SARS-CoV-2 reported infection. Normalized intron excision ratio was obtained from GTEx publicly available sQTL phenotype matrices (<https://www.gtexportal.org/home/datasets>). The genotypes were obtained from whole exome sequence data applied through dbGaP.

-Have the authors considered looking at whether of their splice QTLs with a significant MR causal effect are also protein QTLs in publically available datasets? Both blood and lung splicing could potentially read out as apparent or real altered protein abundance in

plasma/serum (e.g. if an aptamer or antibody targets an epitope that is impacted by alternative splicing). This would add significantly to our understanding of the biological pathways.

This is very informative suggestion. Indeed, in our previous paper¹, we showed that sQTL for *OAS1* (rs10774671) is in high LD ($r^2 = 0.97$) with pQTL for *OAS1* (rs4767027). Thus, *OAS1* levels as measured by the SomaScan platform may reflect p46 isoform. We have now searched for the other five proteins of our interest (*ATP11A*, *DPP9*, *MUC1*, *NPNT*, *PMF1*) in two large pQTL GWASs from deCODE genetics⁵ (N=4,907 proteins of SomaScan assay in 35,559 Icelanders), and from Fenland study⁶ (N= 4,775 proteins measured by the SomaScan and the Olink proximity extension assay in 12,345 individuals, predominantly of White British ancestry).

Although *ATP11A*, *DPP9*, and *PMF1* were not measured in the two datasets, we found two proteins; *MUC1* and *NPNT* that were measured in SomaScan platform. We identified that the sQTL for *NPNT* in lung (rs34712979) also serves as a pQTL for plasma *NPNT* levels. The sQTL for *NPNT*, rs34712979-A allele, creates a cryptic NAGNAG splice acceptor site and is associated with decreased plasma *NPNT* level, both in the deCODE genetics⁵ study and the Fenland study⁶. This suggests that plasma *NPNT* level may act as a read-out of the splicing effect in plasma. However, we acknowledge that we cannot conclude whether this reflects the real or apparent protein abundance caused by the aptamer binding effect impacted by alternative splicing. We have now added the following sentence in the Results, page 7, lines 166-168:

“rs34712979-A allele also serves as a protein-QTL that associates with decreased circulating levels of plasma *NPNT*^{27,28}, which could reflect the aptamer binding effect impacted by the alternative splicing.”

-Did the authors check whether the splice QTLs used as instruments for the significant MR findings show any evidence of pleiotropy in terms of acting on other molecular traits? E.g. are any trans eQTLs?

We have searched for potential pleiotropic effects of the sQTLs used as instruments using Open Targets Genetic (<https://genetics.opentargets.org>). We acknowledge that the sQTLs used as instruments for the significant MR findings also act as eQTLs for other transcripts, which are fully described in Supplementary Table 6.

We have performed MR and colocalization analyses for all genes with eQTL evidence and COVID-19 outcomes (Supplementary Table 7,8). Our sensitivity analysis did not support the causal effects of the global expression of these genes on COVID-19 outcomes, except for *ABO*, *GBAP1*, and *OAS3*. Thus, for those genes, we could not clarify whether the associations with COVID-19 outcomes were driven by either total gene expression or the spliced isoform expression, or both.

Supplementary Table 6: Pleiotropic eQTL effects of the sQTLs used as instruments for the significant MR findings with high colocalization.

EA: Effect Allele, NEA: Non-effect allele

gene	rsid	chr	pos (b38)	EA	NEA	List of other genes for which the sQTL has eQTL effects
MUC1	rs4072037	1	155192276	C	T	THBS3, EFNA1, ADAM15, GBA, MTX1, YY1AP1, DAP3, TRIM46, FAM189B, HCN3, RIT1, DPM3, KRTCAP2, SYT11, SEMA4A, PKLR, FDPS, SCAMP3
GBAP1	rs2974937	1	155199058	C	T	THBS3, EFNA1, ADAM15, GBA, MTX1, YY1AP1, DAP3, TRIM46, FAM189B, HCN3, RIT1, SYT11, DPM3, KRTCAP2, SEMA4A, MUC1, PKLR, SCAMP3
THBS3	rs2066981	1	155202588	A	G	ADAM15, MTX1, YY1AP1, DAP3, TRIM46, RIT1, SYT11, DPM3, KRTCAP2, GBA, SEMA4A
PMF1	rs1052067	1	156236330	C	T	SEMA4A, GLMP, SMG5, PAQR6, TMEME79, SLC25A44, CCT3
NPNT	rs34712979	4	105897896	A	G	-
ABO	rs550057	9	133271182	T	C	GBGT1
OAS1	rs10774671	11	112919388	G	A	OAS3, RPH3A, OAS2
ATP11A	rs12585036	13	112881427	C	T	-
DPP9	rs12610495	19	4717660	A	G	TNFAIP8L1

-Supplementary Excel tables would benefit from more clarity in terms of column headers and explanations – perhaps best as accompanying legends in the manuscript file. E.g. it took me a while to understand “Method/SNP” in ST 1 – which I think shows the position of the instrument if

a Wald test was used but says “All- inverse variance weighted” if multiple instruments were used. I would suggest SNPs in one column with Method in another.

We have revised as suggested. Supplementary Tables 1-3,7,8 now have two columns named “Method” and “Variant list”, which detail the methods used for MR and the list of instrumental variables. Here are the first few lines of Supplementary Table 1, as an example.

exposure	ensembleID	hgnc_symbol	outcome	Method	Variant list	OR	LL	UL	p-value
chr9:133257542:133258097:clu_63938:ENSG00000175164	ENSG00000175164	ABO	Reported infection	Wald ratio	chr9:133271182:C:T	1.15	1.14	1.17	4.04E-78
chr3:46365045:46371244:clu_46934:ENSG00000223552.1	ENSG00000223552	CCR5AS	Critical illness	Wald ratio	chr3:46153097:A:C	2.26	2.04	2.52	2.70E-51
chr19:4714337:4717615:clu_25374:ENSG00000142002.16	ENSG00000142002	DPP9	Critical illness	Wald ratio	chr19:4717660:G:A	0.39	0.35	0.44	3.05E-51
chr3:46365045:46371244:clu_46934:ENSG00000223552.1	ENSG00000223552	CCR5AS	Hospitalization	Wald ratio	chr3:46153097:A:C	1.66	1.55	1.79	1.10E-43
chr19:4714337:4717615:clu_25374:ENSG00000142002.16	ENSG00000142002	DPP9	Hospitalization	Wald ratio	chr19:4717660:G:A	0.56	0.52	0.61	4.17E-41

Similarly, the exposure in ST1 is a paste of a genomic position and gene name. Is the genomic position the TSS, the intron position or the SNP used as the instrument?

We have also clarified the meaning of exposure in the legend as following.

Original: “exposure: excised intron region of the gene”

Edits: “exposure: “chrX:POS1:POS2:clusterID:ensembleID” represents the excised intron region, the clusters each intron belongs to (defined in GTEx), and the gene ensemble ID.”

Minor comments on the Discussion

-The authors identify shared genetic influences on IBD and COVID-19 at the MUC1 locus. It might be worth discussing how it is plausible that altered mucin could impact both COVID and IBD risk, given that IBD is a disease of epithelial surfaces and dysregulated microbiota appears to be involved in its pathogenesis. However, I accept this is speculative and is only a suggestion for the Discussion.

We thank the Reviewer #1 for this sensible suggestion. We have now added the following sentences, in the Result, page 13, lines 327-329.

“The *MUC1* sQTL was also associated with IBD with high colocalization evidence. IBD is a disease with disrupted intestinal epithelial barrier and is suggested to be associated with gut dysbiosis⁵⁶.”

- The authors speculate that targeting gene splicing in COVID19 may be helpful therapeutically. This was interesting but I think unlikely to happen. There are much more tractable targets e.g. inflammatory molecules that are the target of existing drugs. The fact that even these anti-inflammatory therapies have not been taken forward speaks to the uphill battle for a new therapeutic in severe COVID (i.e. it has to demonstrate additional benefit in an RCT above and beyond steroids and IL6R inhibition).

We acknowledge that there are many potentially druggable inflammatory molecules for COVID-19 without evidence of human genetics. We therefore agree with the Reviewer #1 that the targeting alternative splicing only in COVID19 therapy may be challenging to pursue, given that COVID-19 drug development activities have attenuated due to the smaller number of severely affected patients around the world. Nevertheless, our data showed that most of the key alternative splicing in lung may have a shared causal role in other diseases, such as *ATP11A* and *DPP9* for IPF, *NPNT* for COPD. Thus, we believe alternative splicing could be an attractive mechanism for some future drug development opportunities^{7,8}.

-The authors present MR evidence of a causal effect as evidence of a good potential therapeutic target. I'm familiar with this logic and with the data suggesting drugs targeting molecules implicated by human genetics are more likely to be successful. However, I would suggest that MR's value in therapeutic target prioritization may be higher in chronic diseases (cardiovascular disease and LDL-cholesterol being the obvious MR exemplar). Genetics has proven less effective in identifying therapies in COVID19. Anti-IL6R works very well despite lack of any convincing COVID GWAS signal at the well-known common variant that affects IL6R cleavage. Conversely, human genetics implicates type 1 interferon genes and yet interferon therapy has not produced convincing results. I would suggest that in dynamic acute contexts like the host immune response to infection, going after the causal triggers may not work (by analogy, arresting the first rioter won't stop the riot once it's in full swing...).

We understand the Reviewer #1's opinion that targeting the causal triggers sometimes may not work after the dynamic host response begins, which is especially the case for acute diseases. However, we would argue that “Genetics has proven less effective in identifying therapies in COVID19.” is not true. Indeed, genetically proxied IL-6R inhibition had been shown in two studies to reduce the risk of COVID-19 severity in MR framework^{9,10}. Moreover, GWAS for COVID-19 severity has identified that *JAK1* and *TYK2* loci (rs11208552, and rs34536443, respectively) were associated with COVID-19 severity^{11,12}. JAK inhibitors, such as baricitinib and tofacitinib, have been approved for the treatment of COVID-19 in some countries. After we identified that *OAS1* to be an attractive drug target for COVID-19 severity using MR¹, a stimulator

of *OAS1*, interferon lambda, has been shown to decrease risk of severe COVID-19¹³. These are convincing examples that host genetic research could aid drug development for COVID-19.

-The suggestion of bronchoscopically delivered therapy also seems unrealistic. Bronchoscopies are generally avoided in COVID19 (aerosol generating) unless there is another indication such as a suspicion of alternative or concurrent infection or other diagnosis, and therefore this group is not one in whom SSOs would be appealing.

We agree with the Reviewer #1's comment on the indication of bronchoscopy in COVID-19 patients. We have now removed the suggestion of bronchoscopy-delivered therapy.

Reviewer #2 (Remarks to the Author):

This is an interesting paper in which the authors integrate GWAS data from the COVID-19 Host Genetics Initiative with splice QTL data from GTEx to identify genetic variants that influence COVID-19 susceptibility by modulating transcriptional splicing. While the results are interesting and the methods are technically sound, the manuscript would benefit from in depth editing to improve readability and clarify findings. In particular, more information is needed about the sQTLs highlighted as relevant to disease so that inferences can be made as to biological mechanism.

We thank the Reviewer #2 for suggestions to improve our manuscript.

Major Comments

-This manuscript requires significant editing for grammar and English language

We appreciate the Reviewer #2 for this comment. We have tried to edit the manuscript to correct grammatical errors.

-The authors should provide information about the effect size and direction of effect for the sQTLs highlighted in the manuscript (*OAS1*, *ATP11A*, *DPP9*, *NPNT*, and *MUC1*).

We appreciate the Reviewer #2's sensible comments to make our manuscript clearer. We have now added the effect sizes of the sQTLs in the manuscript as following.

1. *OAS1* sQTL, in the Results, page 5, lines 130-132:

"*OAS1* sQTL, rs10774671:A>G, which increases the excision of the intron junction at chr12:112,917,700-112,919,389 [GRCh38] by 1.7 SD per one copy in lung (Fig. 2A, Supplementary Fig. 1A) and by 1.8 SD per one copy in whole blood, respectively,"

2. *ATP11A* sQTL, in the Results, page 6, lines 140-141:

"We additionally identified that *ATP11A* sQTL, rs12585036:T>C, which increases the excision of the intron junction at chr13:112,875,941-112,880,546 by 0.56 SD in lung,"

3. *DPP9* sQTL, in the Results, page 6, lines 150-151:

"We also found novel associations of *DPP9* sQTL, rs12610495:G>A, which increases the excision of the intron junction at chr19:4,714,337-4,717,615 by 0.25 SD in lung,"

4. NPNT sQTL, in the Results, page 6, lines 159-160:

“NPNT sQTL, rs34712979:A>G, which increases the excision of the intron junction at chr4:105,898,001-105,927,336 by 0.64 SD in lung”

5. MUC1 sQTL, in the Results, page 7, lines 178-180:

“We demonstrated the rs4072037-C allele, which increases the excision of the intron junction at chr1:155,192,310:155,192,786 by 1.53 SD in lung,”

It would also be useful to include sashimi plots detailing the splicing events described in the text, and either include quantitative measurements of splice ratios or include box plots of splicing ratios by genotype so the reader can get a visual representation of the magnitude and direction of effect for each sQTL. There is currently insufficient data for the reader to understand what changes are occurring according to genotype and which changes are associated with disease. At the very least there should be a cartoon diagram describing the splice changes that are associated with genotype and disease.

We thank the Reviewer #2' for this important suggestion. We have now applied to dbGAP to get access to the GTEx protected access data⁴ (individual-level genotype and RNA sequencing data) and received the data. We have now provided sashimi plots detailing the splicing events (Supplementary Fig 1.).

Supplementary Fig. 1. Sashimi plots demonstrating the GTEx Leafcutter lung sQTL results.

Sashimi plots to visualize the splice sites using the individual RNA-seq mapped bam files for lung samples in GTEx v.8. Sashimi plots combine the information of read coverage along a gene with curves connecting splice sites supported by RNA-seq data. We selected the maximum equal number of random samples per each sQTL genotype. a) *OAS1* (N=19 for each genotype). b) *ATP11A* (N=12 for each genotype). c) *DPP9* (N=19 for each genotype). This sashimi plot does not demonstrate the sQTL effect of rs12610495 that A-allele was associated with increased excision of the intron junction at chr19:4,714,337-4,717,615 in GTEx. This could be due to selecting random samples of small size (N=19). d) *NPNT* (N=12 for each genotype) e) *MUC1* (N=47 for each genotype), and f) *PMF1* (N=16 for each genotype).

c *DPP9* chr19:4,714,337-4,717,615

d *NPNT* chr4:105,898,001-105,927,336

e *MUC1* chr1:155,192,310-155,192,786

f *PMF1* chr1:156,233,728:156,236,349

In Fig. 2, we have also visualized the effect of the sQTLs on the normalized intron excision ratios.

Fig. 2. The violin plots of normalized intron excision ratio stratified by the sQTL genotypes.

a) *OAS1*, b) *ATP11A*, c) *DPP9* and d) *NPNT* show the sQTLs in dark orange that are associated with COVID-19 severity. e) *MUC1* and f) *PMF1* show the sQTLs in dark blue that are associated with SARS-CoV-2 reported infection. Normalized intron excision ratio was obtained from GTEx publicly available sQTL phenotype matrices (<https://www.gtexportal.org/home/datasets>). The genotypes were obtained from whole exome sequence data applied through dbGaP.

-Leafcutter sQTL results typically include multiple introns grouped into clusters. In this data there only appears to be information for one intron per gene. What happened to the rest of the introns in the cluster? How was the best intron selected? Without information about the individual introns in the cluster it is difficult to characterize what is happening at the splice site.

We thank the Reviewer #2 for this important comment. We have tested individual excised introns per gene without regard to which cluster they belong to. However, we acknowledge that it is important to inform which cluster each intron belongs to. We have thus added the cluster information to the supplementary tables 1-5 (CHR:START:END:ClusterID:EnsembleID in exposure). Here are the first few lines of Supplementary Table 1 as an example.

exposure	ensembleID	hgnc_symbol	outcome	Method	SNPlist	OR	LL	UL	p-value
chr9:133257542:133258097:clu_63938:ENSG00000175164	ENSG00000175164	ABO	Reported infection	Wald ratio	chr9:133271182:C:T	1.15	1.14	1.17	4.04E-78
chr3:46365045:46371244:clu_46934:ENSG00000223552.1	ENSG00000223552	CCR5AS	Critical illness	Wald ratio	chr3:46153097:A:C	2.26	2.04	2.52	2.70E-51
chr19:4714337:4717615:clu_25374:ENSG00000142002.16	ENSG00000142002	DPP9	Critical illness	Wald ratio	chr19:4717660:G:A	0.39	0.35	0.44	3.05E-51
chr3:46365045:46371244:clu_46934:ENSG00000223552.1	ENSG00000223552	CCR5AS	Hospitalization	Wald ratio	chr3:46153097:A:C	1.66	1.55	1.79	1.10E-43
chr19:4714337:4717615:clu_25374:ENSG00000142002.16	ENSG00000142002	DPP9	Hospitalization	Wald ratio	chr19:4717660:G:A	0.56	0.52	0.61	4.17E-41

-Lines 230-231 –The A allele of rs34712979 has also been shown to be associated with increased COPD risk (not just FEV1/FVC ratio) through an sQTL in lung tissue – please site PMID 33173926

Although we have already cited this paper (ref 26) in page 7, line 166, we have also cited this paper in page 10, lines 249-250 as following.

“The NPNT sQTL had similar trend for the rs34712979-A allele, which was protective for COVID-19 severity, was associated with increased risk of COPD²⁶”

Minor Comments

-Line 128-131- please specify the variant ID and the direction of effect so that it is not necessary to refer to your previous paper to understand this result

We have revised the sentence in the Results, on page 5, lines 128-135 as following,

“We first replicated the association of COVID-19 outcomes with OAS1 splicing, using an updated version of COVID-19 HGI release 7, which provides a 10-fold increase in case sample size. OAS1 sQTL, rs10774671:A>G, which increases the excision of the intron junction at chr12:112,917,700-112,919,389 [GRCh38] by 1.7 SD per one copy in lung (Fig. 2A, Supplementary Fig. 1A) and by 1.8 SD per one copy in whole blood, respectively, was associated with protection against all three adverse COVID-19 outcomes. The higher excision of the intron junction at chr12:112,917,700-112,919,389 corresponds to an increased level of the p46 isoform, a prenylated form of OAS1 with higher anti-viral activity than the p42 isoform^{18,19,25}”

-The authors should define a new term or abbreviation only once in the main text of the manuscript – for example sQTL is defined on lines 72, 81-82, 95, 364;

We have modified accordingly.

-Instead of the term “gene splicing” – “RNA” or “transcriptional” splicing would be more accurate

We thank the Reviewer #2 for correcting our terminology. We have now avoided the term “gene splicing” to be more accurate. We have now used either “splicing”, “RNA splicing”, “alternative splicing” or “splicing events” throughout the manuscript.

-lines 106-107: “A total of 4,477 genes in lungs ... contained conditionally independent cis-sQTLs that were also present in the GWAS meta-analyses...” This statement is unclear – do the authors mean that 4,477 genes in lung tissue, and 2,779 genes in whole blood contained splice sites that were associated with at least one SNP that was also tested in the GWAS? It would be helpful to clarify how many splice sites, genes, sQTL-SNPs and sQTL SNP-gene pairs are being referred to here (ideally in a table in the main text)

Although the requested information is shown in Fig 1, we admit that the statement in the main text is unclear. We therefore modified the sentences in the Results, on Page 4, lines 109-111:

“A total of 5,724 transcriptional splicing events for 4,329 genes in lung and 3,568 transcriptional splicing events for 2,671 genes in whole blood contained conditionally independent cis-sQTLs that were also present the GWAS meta-analyses”

-lines 277-285 (description of splice-switching oligonucleotides) should be moved towards the end of the discussion, the current position does not flow logically.

We appreciate the Reviewer #2 for improving our manuscript. We have now moved this paragraph to the end of the manuscript (lines 360-370), which now flows more logically better than before.

References

1. Zhou, S. *et al.* A Neanderthal OAS1 isoform protects individuals of European ancestry against COVID-19 susceptibility and severity. *Nat. Med.* (2021) doi:10.1038/s41591-021-01281-1.
2. Machiela, M. J. & Chanock, S. J. LDlink: a web-based application for exploring population-specific haplotype structure and linking correlated alleles of possible functional variants. *Bioinformatics* **31**, 3555–3557 (2015).
3. Bonnevie-Nielsen, V. *et al.* Variation in antiviral 2',5'-oligoadenylate synthetase (2'5'AS) enzyme activity is controlled by a single-nucleotide polymorphism at a splice-acceptor site in the OAS1 gene. *Am. J. Hum. Genet.* **76**, 623–633 (2005).
4. GTEx Portal. <https://www.gtexportal.org/home/protectedDataAccess>.
5. Ferkingstad, E. *et al.* Large-scale integration of the plasma proteome with genetics and disease. *Nat. Genet.* (2021) doi:10.1038/s41588-021-00978-w.
6. Pietzner, M. *et al.* Synergistic insights into human health from aptamer- and antibody-based proteomic profiling. *Nat. Commun.* **12**, 6822 (2021).
7. Schneider-Poetsch, T., Chhipi-Shrestha, J. K. & Yoshida, M. Splicing modulators: on the way from nature to clinic. *J. Antibiot. (Tokyo)* **74**, 603–616 (2021).
8. Le, K., Prabhakar, B. S., Hong, W. & Li, L. Alternative splicing as a biomarker and potential target for drug discovery. *Acta Pharmacol. Sin.* **36**, 1212–1218 (2015).
9. Larsson, S. C., Burgess, S. & Gill, D. Genetically proxied interleukin-6 receptor inhibition: opposing associations with COVID-19 and pneumonia. *Eur. Respir. J.* **57**, (2021).
10. Bovijn, J., Lindgren, C. M. & Holmes, M. V. Genetic variants mimicking therapeutic

inhibition of IL-6 receptor signaling and risk of COVID-19. *Lancet Rheumatol.* **2**, e658–e659 (2020).

11. Pairo-castineira, E. *et al.* Genetic mechanisms of critical illness in Covid-19. *Nature* **17**, 25 (2020).
12. COVID-19 Host Genetics Initiative. <https://www.covid19hg.org/results/r7/>.
13. Reis, G. *et al.* Early Treatment with Pegylated Interferon Lambda for Covid-19. *N. Engl. J. Med.* **388**, 518–528 (2023).

REVIEWER COMMENTS

Reviewer #1 (Remarks to the Author):

I thank the authors for their comprehensive and thoughtful responses. My comments have all been addressed.

Reviewer #2 (Remarks to the Author):

The manuscript has significantly benefited from edits for improved readability and clearer presentation of data. However, the sashimi plots do not have me convinced that there is a true difference in splicing associated with genotype for several of the genes. This could be improved by clearer sashimi plots. Please show the exon coverage of the exons in the transcript. Also it would help to only show the junctional reads that are involved in the splice event that the authors are trying to demonstrate. In addition, there should be more description of the splicing event that is happening in each gene either in the figure legend or the main text.

We thank the Reviewers and Editors for their constructive feedback and the invitation to re-submit our work to *Nature Communications*. Below we address, point-by-point, the comments of the Reviewers. Our responses are in **blue font** and modified manuscript text are in **orange font** to improve readability and all changes to the manuscript have been denoted by line numbers.

REVIEWER COMMENTS

Reviewer #1 (Remarks to the Author):

I thank the authors for their comprehensive and thoughtful responses. My comments have all been addressed.

We thank again the Reviewer #1 for his/her supportive comments.

Reviewer #2 (Remarks to the Author):

The manuscript has significantly benefited from edits for improved readability and clearer presentation of data. However, the sashimi plots do not have me convinced that there is a true difference in splicing associated with genotype for several of the genes. This could be improved by clearer sashimi plots. Please show the exon coverage of the exons in the transcript. Also it would help to only show the junctional reads that are involved in the splice event that the authors are trying to demonstrate. In addition, there should be more description of the splicing event that is happening in each gene either in the figure legend or the main text.

We appreciate the Reviewer #2's comments to improve the presentation of sashimi plots. As suggested, the sashimi plots (Supplementary Fig. 1) only show the junctional reads that are involved in the splice events. We have also adjusted the junctional reads for the coverage of the regions using counts per million (CPM), and added more descriptions for the splicing event of each gene in the legend.

Supplementary Fig. 1. Sashimi plots demonstrating the GTEx Leafcutter lung sQTL results.

Sashimi plots to visualize the splice sites using the individual RNA-seq mapped bam files for lung samples ($N=514$) in GTEx v.8. Sashimi plots combine the information of read coverage along a gene with curves connecting splice sites supported by RNA-seq data. The mean number of reads supporting the splicing events per each genotype group are shown in the sashimi plots, which were adjusted for the average expression (counts per million: CPM) of the region including the cluster to which the index intronic junction belongs to and the exons at both ends. CPM was calculated by the mapped read counts of the region / the total read counts $\times 10^{-6}$ (Supplementary Table 9). The transcripts for all genes except for OAS1 were annotated using gencode.v43.chr_patch_hapl_scaff.annotation.gtf (downloaded from <https://www.gencodegenes.org/human/>). For OAS1, we show only basic transcripts by using gencode.v43.basic.annotation.gtf to reduce the complexity of the figure.

- a) For the OAS1 splicing event, 65 samples with a GG genotype, 255 samples with a GA genotype, and 194 samples with a AA genotype at rs10774671 were used to visualize the splicing event. These data show that the rs10774671-G allele is associated with increased excision of the intron junction at chr12:112,917,700-112,919,389.
- b) For the ATP11A splicing event, 316 samples with a CC genotype, 177 samples with a CT genotype, and 21 samples with a TT genotype of rs12585036 were used to visualize the splicing event. These data show that the rs12585036-C allele is associated with increased excision of the intron junction at chr13:112,875,941-112,880,546.

c **DPP9** chr19:4,714,337-4,717,615

d **NPNT** chr4:105,898,001-105,927,336

- c) For the DPP9 splicing event, 278 samples with a AA genotype, 191 samples with a AG genotype, and 45 samples with a GG genotype at rs12610495 were used to visualize the splicing event. These data show that the rs12610495-A allele was associated with increased excision of the intron junction at chr19:4,714,337-4,717,615.

d) For the NPNT splicing event, 33 samples with a AA genotype, 162 samples with a AG genotype, and 319 samples with a GG genotype at rs34712979 were used to visualize the splicing event. These data show that the rs34712979-G allele is associated with increased excision of the intron junction at chr4:105,898,001-105,927,336.

e) For the MUC1 splicing event, 156 samples with a TT genotype, 251 samples with a TC genotype, and 107 samples with a CC genotype of rs4072037 were used to visualize the splicing event. These data show that the rs4072037-C allele is associated with increased excision of the intron junction at chr1:155,192,310-155,192,786.

f) For the PMF1 splicing event, 37 samples with a AA genotype, 208 samples with a AG genotype, and 269 samples with a GG genotype of rs1052067 were used to visualize the splicing event. These data show that the rs1052067-A allele is associated with increased excision of the intron junction at chr1:156,233,728-156,236,349.

REVIEWERS' COMMENTS

Reviewer #2 (Remarks to the Author):

The sashimi plots are much easier to understand and clearly show the splice events in questions. All of my comments have now been addressed.

We thank the Reviewers and Editors for their constructive feedback and the invitation to re-submit our work to *Nature Communications*. Below we address, point-by-point, the comments of the Reviewers. Our responses are in **blue font** to improve readability and all changes to the manuscript have been denoted by line numbers.

REVIEWERS' COMMENTS

Reviewer #2 (Remarks to the Author):

The sashimi plots are much easier to understand and clearly show the splice events in questions. All of my comments have now been addressed.

We thank again the Reviewer 2's comments to improve the presentation of sashimi plots and we now appreciated his/her supportive comments.